# ScaleCom: Scalable Sparsified Gradient Compression for Communication-Efficient Distributed Training

**Chia-Yu Chen[1], Jiamin Ni[2], Songtao Lu[2], Xiaodong Cui[1], Pin-Yu Chen[2]**
**Xiao Sun[1], Naigang Wang[1], Swagath Venkataramani[2]**
**Vijayalakshmi Srinivasan[1], Wei Zhang[1], Kailash Gopalakrishnan[1]**
IBM T. J. Watson Research Center
Yorktown Heights, NY 10598, USA
[1]{cchen, cuix, xsun, nwang, viji, weiz, kailash}@us.ibm.com
[2]{jiamin.ni, songtao, pin-yu.chen, swagath.venkataramani}@ibm.com

## Abstract

Large-scale distributed training of Deep Neural Networks (DNNs) on state-of-the-art platforms is expected to be severely communication constrained. To overcome this limitation, numerous gradient compression techniques have been proposed and have demonstrated high compression ratios. However, most existing methods do not scale well to large scale distributed systems (due to gradient build-up) and/or fail to evaluate model fidelity (test accuracy) on large datasets. To mitigate these issues, we propose a new compression technique, Scalable Sparsified Gradient Compression (ScaleCom), that leverages similarity in the gradient distribution amongst learners to provide significantly improved scalability. Using theoretical analysis, we show that ScaleCom provides favorable convergence guarantees and is compatible with gradient all-reduce techniques. Furthermore, we experimentally demonstrate that ScaleCom has small overheads, directly reduces gradient traffic and provides high compression rates (65-400X) and excellent scalability (up to 64 learners and 8-12X larger batch sizes over standard training) across a wide range of applications (image, language, and speech) without significant accuracy loss.

## 1 Introduction

Over the past decade, DNNs have surpassed traditional Machine Learning models on a wide range of applications including computer vision [1][2], speech [3], and natural language processing (NLP) [4][5]. As models and datasets have grown in complexity, training times have increased significantly [2][4]. To tackle this challenge, data-parallelism approaches are widely used to accelerate the training of DNN models [6]. In order to scale data-parallelism techniques to more workers while preserving the computational efficiency in each worker, it is important to increase the overall batch size proportionally with the number of workers. However, increasing the batch size often leads to a significant loss in test accuracy–remedied by a number of recent ideas including increasing the learning rate during the training process as well as a learning rate warm-up procedure [7][8][9]. Using these techniques, large batch size training has been successfully applied to state-of-the-art distributed systems [10][11]. However, increasing evidence seems to suggest that there is a maximum mini-batch size beyond which the number of iterations required to converge increases [12]. Furthermore, driven by recent advances in low-precision arithmetic [13][14][15], there has been a renaissance in the computational capability of deep learning training hardware resulting in accelerator throughputs exceeding 100s of TeraOps/s [16][17][18][19]. This dramatic increase in throughput can cause an imbalance between computation and communication, resulting in large scale training platforms that are severely communication constrained.

To mitigate these communication bottlenecks in DNN training, several gradient compression techniques have been proposed [20][21][22][23]. Most of these techniques exploit error feedback or 'local memory' (preserving gradient residues from compression) to demonstrate significant compression rates and good convergence properties. However, current error-feedback gradient compression techniques cannot be directly applied to large-scale distributed training. There are two primary challenges. (a) **Gradient build-up**: As addressed in [24][25][26][27], compressed data can be gathered, but not reduced. This results in a dramatically decreased compression rate as the number of workers increases. (b) **Large batch size with scaled learning rate**: As shown in [28], for a convex problem, the noise term in the error-feedback gradient increases as the cube of the learning rate ($\alpha^3$). [29] also shows that the increased learning rate could add large noise for error-feedback gradient compression in non-convex and distributed settings. Thus, scaled learning rates needed for large batch-sized training can significantly increase gradient noise and cause performance degradation (or even divergence), particularly for complex models and datasets.

In this paper, we propose a new gradient compression algorithm, ScaleCom, that provides solutions to both of these challenges. ScaleCom provides significant compression rates (65-400X) while enabling convergence in large-scale distributed training (64 workers). To the best of our knowledge, this is the first compression algorithm that has been extensively evaluated in large datasets and batch sizes and shown to be fully compatible with conventional all-reduce schemes, as shown in Table 1.

Table 1: Comparing different compressors for error-feedback SGD

| Compressor | scalability | overhead (FLOPs/element) | compr. rate | convergence | empirical exp. | LB[e] |
|---|---|---|---|---|---|---|
| Top K[21][30] | $\mathcal{O}(n)$ | $\mathcal{O}(log\ p)$ (sort)[a] | >100X | not guaranteed[b] | broadly tested | no |
| AdaComp[22] | $\mathcal{O}(n)$ | $\sim 4$ (quasi-sort) | 40-200X | not guaranteed | broadly tested | no |
| DGC[23] | $\mathcal{O}(n)$ | $\mathcal{O}(1)$ (sample based-sort) | 270-600X | not guaranteed | broadly tested | no |
| PowerSGD[26] | $\mathcal{O}(\log(n))$ | low-rank approximation | 40-128X | not guaranteed | small datasets | yes |
| gTop-k[27] | $\mathcal{O}(\log(n))$ | local top-$k$ merge | >100X | not guaranteed | up to 6% degrad. | no |
| SketchSGD[24] | constant | $2 * H(.) * r$ (sketch table)[c] | 40X | guaranteed | transformer | no |
| **ScaleCom (ours)** | **constant** | $\sim 3$ **(chunk-wise sort)** | **65-400X** | **guaranteed** | **broadly tested**[d] | **yes** |

[a] $p$ is mode size. [b] unless explicit assumption is made. [c] H(.) is hash function computation and r is rows of sketch table. [d] include a wide range of applications with large datasets. [e] large batch size training/scaled learning rate.

## 1.1 Challenges and Related Works

**Error-feedback gradient compression and all-reduce:** Error-feedback gradient compression was first introduced by [20] and later widely applied to various application domains [21][22][23][30]. Error-feedback gradient (also referred to as "residues" [22] or local memory) is the difference between a worker's computed gradient and it's compressed gradient. When compressed gradients from multiple workers are sent to a centralized parameter server (for reduction), they cause a "gradient build-up" problem. Specifically, as shown in Figure 1(a), since different workers pick different gradients during compression, the overall compression ratio for the accumulated gradients decreases linearly with the number of workers $n$, i.e., $\mathcal{O}(n)$. This effect is especially dramatic in large-scale distributed systems as shown in Figure 1(b). Recently, there has been a body of work focused on the gradient build-up issue. [25] emphasizes the importance of commutability in gradient compression to enable efficient aggregation in ring all-reduce. [31] proposed low-rank methods for error-feedback gradient compression that reduces the complexity to $\mathcal{O}(\log n)$. [24] used the reduction property of sketch tables to achieve 40X compression rates. [32] did *double compression* to achieve linear speedup. [27] merged each worker's top elements to approximate the all-reduce of global top elements. In spite of all these efforts, none of these techniques have been shown to comprehensively work on large models, datasets and high number of learners, with the desired $\mathcal{O}(1)$ constant complexity.

**Large batch size training:** Furthermore, many of these compression techniques have not shown to work well in large batch size training scenarios where communication bottlenecks limit system performance and scalability. [22] and [23] scaled mini-batch sizes by 8X and achieved baseline accuracies for CIFAR10 models. Similarly, [31] linearly scaled the learning rate and batch size by 16X and reduced communication time by 54% in ResNet18 (CIFAR10). Overall, most recent studies have primarily focused on small datasets, and it remains unclear if gradient compression techniques work well on large models and datasets. As shown in Figure 1(c), we observe that a naive error-feedback gradient compression [21] scheme can cause significant accuracy degradation in large batch size training scenarios (Transformer in WMT14 En-De).

**Convergence analyses of error-feedback gradient compression:** In addition to empirical results, [28] and [29] provided convergence analyses for error-feedback gradient compression in both convex and non-convex optimization contexts and show convergence similar to traditional stochastic gradient descent (SGD). The results suggest that the essence of network convergence is the contraction property of compressors, defined as the "energy" preserved in the compressed gradients relative to the full gradients as shown in Eqn.(4) of [28]. The results show that both random-$k$ and top-$k$ compression could achieve similar convergence properties as SGD. Later on [33] reported the advantages of the top-$k$ compressor. Recent analyses [34] also proved that error feedback can enable biased gradient compressors to reach the target test accuracy with high compression rates. In theory, compressors are quite flexible (biased or unbiased).

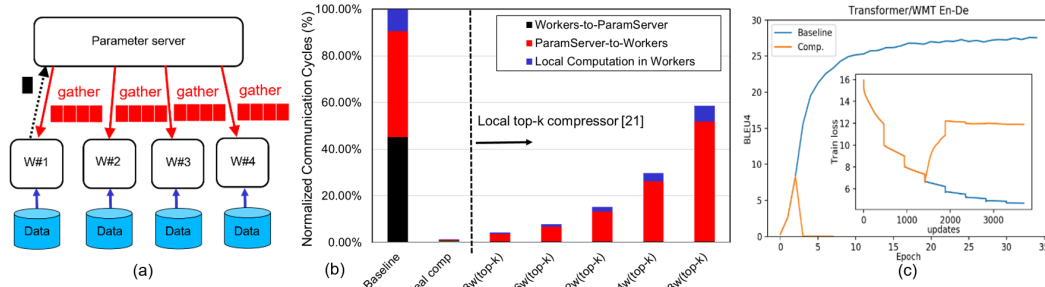

Figure 1: Challenges for gradient compression in large batch size training:(a) Illustration of 'gradient build-up' issue for compressed gradients. Compressed gradients cannot be reduced directly; instead they are gathered. Gather operation does not scale well to worker number (red). (b) Communication bottlenecks due to gradient build-up; as worker number increases, communication from parameter server to workers becomes a server bottleneck. In this experiment, ResNet50 (ImageNet), bandwidth=32GBps, and compression rate 112X are used. Performance model is based on [35]. (c) In large batch size training, standard local top-$k$ gradient compression [21] could cause model divergence: Transformer in WMT14 En-De for 288k batch size with 64 workers.

## 1.2 Contributions

In this paper, we introduce a new gradient compression technique, ScaleCom, that resolves the two important issues central to scalability: (i) enable compression to work effectively with all-reduce and (ii) applicable to large batch size training for large datasets. In comparison to existing compression methods, our primary contributions include:

1. We explore local memory (error feedback) similarity across workers and use this property to design a commutative compressor, which we call cyclic local top-$k$ (CLT-$k$). The CLT-$k$ operator solves the gather (gradient build-up) issue and is compatible with all-reduce operations.

2. To apply gradient compression in large batch size training, we propose a novel low-pass filter during local memory updates. This filter cleans out disruptive noise and enhances local memory similarity. Thus, our filter scales the CLT-$k$ compressor to much larger-scale distributed training.

3. We present theoretical analysis to show that ScaleCom can guarantee the same convergence rate as SGD and enjoys linear speedup with the number of workers. ScaleCom mitigates gradient noise induced by scaled learning rates and keeps communication cost constant with the number of workers. Moreover, we have also observed that ScaleCom has similar convergence properties as the ideal (but impractical) *true top-$k$* compression.

4. Experimentally, we have verified that ScaleCom shows no degradation across a wide range of applications (datasets) including vision (ImageNet), language (WMT), and speech (SWB300), in both standard (8 workers) and large batch size (64 workers) training.

## 2 Gradient Sparsification in All-Reduce

A commutative compressor between gradient averaging and sparsification following definition (1) is desired for communication-efficient distributed training. There are two advantages for commutative compressors: (i) theoretically, with this setting, error-feedback gradient compression has convergence guarantees [29], and (ii) this resolves the 'gradient build-up' issue and keeps communication cost constant with the number of workers [25].

$$\text{sparse}\left(\frac{1}{n}\sum_{i=1}^{n}\mathbf{x}_i\right) = \frac{1}{n}\sum_{i=1}^{n}\text{sparse}(\mathbf{x}_i) \tag{1}$$

Besides commutativeness, recent studies [23][28][29][33] suggest that the top-$k$ compressor has good contraction properties and test accuracies from both theoretical and empirical perspectives. Thus, an optimized compressor should have both (i) commutative property and (ii) top-$k$ contraction property. To satisfy these, we designed our compressor based on the following two observations:

*(i) Memory similarity:* Although local memory (gradient residue) is never exchanged amongst workers, it is correlated in the sense that local gradients are computed from samples drawn from the same training set. Figure 2(a) shows the pairwise cosine distance (worker 0 and 1)[1] of local memory in the first 90 iterations of ResNet18 (CIFAR10) with conventional local top-k compressor (top-0.1% is used)[21]. The cosine distance decreases fast over the iterations, i.e., local memory similarity is improved quickly and stays correlated over much of the training process. (Appendix-A shows different statistical metrics.) Finally, we observe that this phenomenon is agnostic to increasing worker number when learning rate and per-worker batch size stays the same as shown in Figure 2(a).

*(ii) True vs local top-k:* The local memory similarity amongst workers offers a critical insight: *the local worker's top-k indices may be used to approximate the true top-k indices.* In Figure 2(b), area under blue curve represents *all-reduced* error-feedback gradient magnitudes[2], among which, the area to the right of grey line corresponds to its top $k$ (i.e. *true top-k*).[3] The *true top-k* area overlaps more than 70% with the red histogram representing *local top-k* of worker 0, suggesting that *true top-k* and *local top-k* have sufficiently overlapping indices and similar contraction properties.

**Cyclic Local Top-$k$ (CLT-$k$) Compressor:** Based on the similarity between local memories, we propose a novel efficient commutative compressor for all-reduce distributed training, *cyclic local top-k (CLT-k)*. It works as follows: In each iteration, we sequentially select a leading worker in a cyclical order. The leading worker sorts its error-feedback gradient and obtains its local top-$k$ indices. All other workers follow the leading worker's top-$k$ index selection for compressing their own local error-feedback gradients. Formally, CLT-$k$ compressor is described as follows.

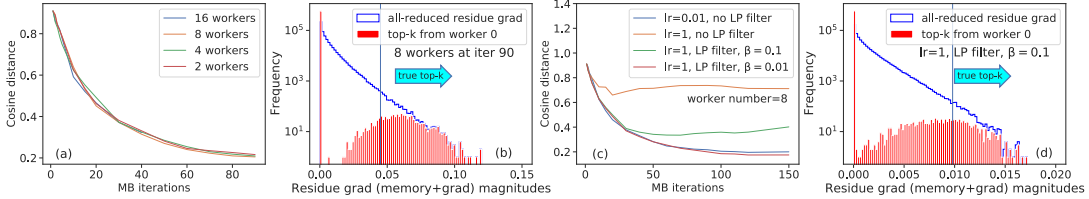

Figure 2: Similarity analysis of error-feedback gradient compression on ResNet18 (CIFAR10): (a) Cosine distance between workers' memories over iterations; (b) Histogram (in log scale) of element-wise residual gradient magnitude at iteration 90 in epoch 0. (c) Cosine distance between workers' memories with varying learning rate and low pass filter's $\beta$ in CLT-$k$. (d) Histogram (in log scale) of element-wise residual gradient magnitude at iteration 90 in epoch 0 with scaled learning rate ($\alpha$=1) and low-pass filter ($\beta$=0.1).

Let $\mathcal{I}^k(\mathbf{x}_i)$ denote the index set corresponding to the indices of the $k$ largest entries (in magnitude) of vector $\mathbf{x}_i$. To be more specific, the set is defined by

$$\mathcal{I}^k(\mathbf{x}_i) = \{m : |(\mathbf{x}_i)_m| \geq |(\mathbf{x}_i)_k|; \; |(\mathbf{x}_i)_k| \text{ is the } k\text{th largest entry in magnitude of } \mathbf{x}_i\} \quad (2)$$

Suppose that there are $n$ vectors $\{\mathbf{x}_i\}_{i=1}^n$ Then, we have $n$ local top-$k$ sets, i.e., $\{\mathcal{I}^k(\mathbf{x}_i)\}_{i=1}^n$. For a vector $\mathbf{x}_j$, the proposed CLT-$k$ compressor with worker $i$ as the leader, denoted by $\text{CLT}_i^k : \mathbb{R}^d \to \mathbb{R}^d$, is defined entry-wise as

$$[\text{CLT}_i^k(\mathbf{x}_j)]_m = \begin{cases} (\mathbf{x}_j)_m, & \text{if} \quad m \in \mathcal{I}^k(\mathbf{x}_i) \\ 0, & \text{otherwise.} \end{cases} \quad (3)$$

*Remark 1.* Note that when $i = j$, $\text{CLT}_i^k(\mathbf{x}_j)$ is reduced to the classical top-$k$ operator on $\mathbf{x}_i$. When $i \neq j$, $\text{CLT}_i^k(\mathbf{x}_j)$ sets $\mathbf{x}_j$'s entries whose indices belong to $\mathcal{I}^k(\mathbf{x}_i)$ as 0.

*Remark 2.* It is easy to verify that (3) satisfies the commutative property in (1). Moreover, Figure 2(b) suggests the histogram of error-feedback gradient of $\text{CLT}_i^k(\mathbf{x}_j)$ highly overlaps with that of top-$k$

$(\frac{1}{n}\sum_{j=1}^{n} \mathbf{x}_j)$. Thus, proposed CLT-$k$ compressor features efficient implementation in all-reduce, has desirable commutative properties and shares a similar contraction property with true top-$k$.

*Remark 3.* We note that the proposed compressor can naturally be extended to ring all-reduce settings.

**Low-Pass Filtering in Memory Accumulation:** Large batch size training schemes usually require to significantly scale up learning rate. As shown in Figure 2(c), when learning rate is increased from 0.01 to 1 (100X), cosine distance becomes much larger (orange line), suggesting drastically reduced local memory similarity, which may degrade the performance of the CLT-$k$ compressor. Besides, scaled learning rate causes rapid model changes and incurs larger gradient noise, which makes it more difficult to compress gradients in large batch size settings. To address these challenges, we propose to apply low-pass filtering [36] to local memory accumulation. This low-pass filtering is one kind of weighted error feedback techniques [37][38], but it focuses on large batch size training and aims to mitigate noise from the *incoming residual gradients*. Our filter passes the signals of computed gradients with smoother changes and attenuates the gradient noise caused by rapid model changes, which (i) mitigates undesirable noise caused by scaled learning rate, and (ii) improves local memory similarity among workers. Formally our method is described as follow. Assuming $n$ workers, the distributed training problem is

$$\min_{\boldsymbol{\theta} \in \mathbb{R}^p} f(\boldsymbol{\theta}) := \frac{1}{n} \sum_{i=1}^{n} f_i(\boldsymbol{\theta}) \tag{4}$$

where $f_i(\boldsymbol{\theta}) = \mathbb{E}_{\xi_i \sim \mathcal{D}} F(\boldsymbol{\theta}, \xi_i)$ denotes the objective function at the $i$th worker, $\boldsymbol{\theta}$ is the optimization variable (weights of the neural net), $\xi_i$ represents the sample at node $i$, $\mathcal{D}$ stands for the data distribution. This work focuses on fully-synchronized distributed training so data distributions at different nodes are identical. Let $\mathcal{B}_i$ denote the mini-batch at the $i$th worker; gradient estimate is written as $\widehat{\nabla}_{\boldsymbol{\theta}} f_{\mathcal{B}_i}(\boldsymbol{\theta}) = |\mathcal{B}_i|^{-1} \sum_{j \in \mathcal{B}_i} \nabla_{\boldsymbol{\theta}} f_{i,j}(\boldsymbol{\theta})$, where $\nabla_{\boldsymbol{\theta}} f_{i,j}(\boldsymbol{\theta})$ denotes the gradient of loss function $f_i(\boldsymbol{\theta})$ w.r.t. the $j$th sample at node $i$, and $|\mathcal{B}_i|$ is the batch size of the sampled data at the $i$th worker. Here we use $\mathbf{m}_i$ as gradient residues (local memory) in the $i$th worker and $\mathbf{g}_i$ as the compressed gradient after scaling by step size $\alpha$. These parameters are computed locally, where $\mathbf{g}_i$ will be sent to update shared weight $\mathbf{x}$. Then, the low-pass filter on memory can be written as

$$\mathbf{m}_i^{t+1} = (1 - \beta)\mathbf{m}_i^t + \beta(\mathbf{m}_i^t + \widehat{\nabla}_{\boldsymbol{\theta}} f_{\mathcal{B}_i}(\boldsymbol{\theta}^t) - \mathbf{g}_i^t) \tag{5}$$

where $\beta$ is the discounting factor ($0 < \beta \leq 1$), and $t$ is the number of iterations. Empirically, we verify that the use of low-pass filters can improve the similarity among local memory for CLT-$k$ in the case of scaled learning rate as shown in green and red lines in Figure 2 (c). Figure 2 (d) shows that when the learning rate is significantly increased (100X), with the use of the low-pass filter, our CLT-$k$ compressor can still maintain sufficient area overlap in the histograms with true top-$k$ compressors, providing a necessary and desirable contraction property for robust and scalable training. One thing should be noted that intuitively, this filtering method has a connection to momentum SGD: momentum SGD can be viewed as a form of filtering (moving average) on current and past gradients, which smooths out noisy gradients to update weight more accurately. Analogously, we perform filtering on the residual gradients to improve signal integrity in local memory.

---

**Algorithm 1** ScaleCom: Scalable Sparsified Gradient Compression

---

1: **Input:** initialize shared variable $\boldsymbol{\theta}$ and $\mathbf{m}_i^t = 0, \forall i$
2: **for** $t = 1, \dots, T$ **do**
3:     **for** $i = 1, \dots, n$ in parallel **do**
4:         Select $\mathcal{B}_i$                                                     ▷ set up mini-batch
5:         Compute a stochastic gradient $\widehat{\nabla}_{\boldsymbol{\theta}} f_{\mathcal{B}_i}(\boldsymbol{\theta}^t))$         ▷ each worker computes gradients
6:         $\mathbf{g}_i^t = \text{CLT}_{\text{mod}(t,n)}^k(\mathbf{m}_i^t + \widehat{\nabla}_{\boldsymbol{\theta}} f_{\mathcal{B}_i}(\boldsymbol{\theta}^t))$         ▷ CLT-$k$ compression (3)
7:         $\mathbf{m}_i^{t+1} = (1 - \beta)\mathbf{m}_i^t + \beta\left(\mathbf{m}_i^t + \widehat{\nabla}_{\boldsymbol{\theta}} f_{\mathcal{B}_i}(\boldsymbol{\theta}^t) - \mathbf{g}_i^t\right)$     ▷ low-pass filtering (5)
8:     **end for**
9:     Upload $\{\mathbf{g}_i^t\}$ to the server                          ▷ comm. from workers to parameter
10:     $\mathbf{g}^t = \frac{1}{n}\sum_{i=1}^{n} \mathbf{g}_i^t$                                   ▷ gradient reduction
11:     Download $\{\mathbf{g}^t\}$ to the each worker               ▷ comm. from parameter-server to workers
12:     $\boldsymbol{\theta}^{t+1} = \boldsymbol{\theta}^t - \alpha\mathbf{g}^t$
13: **end for**

---

# 3 Scalable Sparsified Gradient Compression (ScaleCom)

In this section, we will describe the details of our algorithm, ScaleCom, and its convergence properties. In ScaleCom, each worker first applies the CLT-$k$ compressor as shown in (3). Sparsified data is directly added (reduced) across workers (integrated with all-reduce) avoiding 'gradient build-up'. After all-reduce, each worker applies a low-pass filter in local gradient accumulation, improves workers' memory similarity and smooths out abrupt noise induced by scaled learning rates. For simplicity, we used the parameter server protocol to explain our algorithm, but it can naturally be extended to all-reduce ring implementations. The whole process is summarized in Algorithm 1. [4] In the rests of this section, we provide formal convergence properties for ScaleCom. [5]

**Contraction Property:** We establish the contraction property of the CLT-$k$ compressor based on the Hamming distance. The Hamming distance measures the overlap of the two index sets. Suppose $\mathcal{I}^k$ is a set of $k$ indices of a vector $\mathbf{x}$. Define a binarized vector $\mathbf{x}_{\mathcal{I}^k}$ as the following: $\mathbf{x}_{\mathcal{I}^k, m} = 1$, if $m \in \mathcal{I}^k$, otherwise, $\mathbf{x}_{\mathcal{I}^k, m} = 0$. Suppose $\mathcal{I}_1^k$ and $\mathcal{I}_2^k$ are two sets of $k$ indices. The Hamming distance between the two sets given a vector $\mathbf{x}$ and an auxiliary variable $d$ are defined as:

$$\mathbf{H}(\mathcal{I}_1^k, \mathcal{I}_2^k) \triangleq \mathbf{H}(\mathbf{x}_{\mathcal{I}_1^k}, \mathbf{x}_{\mathcal{I}_2^k}) = 2d, \quad 0 \leq d \leq k. \tag{6}$$

**Lemma 1.** *Suppose $\mathbf{y}$ is a vector and its top-k index set is $\mathcal{I}^k$. $\mathbf{y}$ is sparsified by another index set $\tilde{\mathcal{I}}^k$. If $\mathbf{H}(\mathcal{I}^k, \tilde{\mathcal{I}}^k) = 2d$, we have the following contraction property for this compressor $comp(\mathbf{y})$ : $\mathbb{E} \|\mathbf{y} - comp(\mathbf{y})\|^2 \leq \gamma \|\mathbf{y}\|^2$, where*

$$\gamma \triangleq \frac{d}{k} + \left(1 - \frac{d}{k}\right) \cdot \gamma_0 \tag{7}$$

*and $\gamma_0$ is the contraction coefficient of top-k sparsification $\mathbb{E} \|\mathbf{y} - top_k(\mathbf{y})\|^2 \leq \gamma_0 \|\mathbf{y}\|^2$ .*

We can see that depending on $\frac{d}{k}$, the contraction coefficient $\gamma \in [\gamma_0, 1]$. Specialized to the proposed CLT-$k$ compressor, for each iteration $t$ an index set is generated from a local worker $i$ in a cyclic fashion. Let $\mathbf{y} = \frac{1}{n} \sum_{j=1}^{n} (\mathbf{m}_j^t + \widehat{\nabla}_{\boldsymbol{\theta}} f_{\mathcal{B}_j}(\boldsymbol{\theta}^t))$ which is the averaged error-feedback gradients among all workers. We assume $d \leq d_0 < k$ which indicates there exists a minimal overlap $k - d_0$ between the local top-$k$ indices from worker $i$ and global true top-$k$ given by $\mathbf{y}$. Therefore,

$$\gamma \leq \frac{d_0}{k} + \left(1 - \frac{d_0}{k}\right) \cdot \gamma_0 < 1. \tag{8}$$

It follows that $\mathbb{E} \|\mathbf{y} - \text{CLT}_i^k(\mathbf{y})\|^2 \leq \gamma \|\mathbf{y}\|^2$.

**Convergence Analysis:** Before showing the theoretical results, we make the following assumptions.

**A. 1** We suppose that the size of gradient is upper bounded, i.e., $\|\nabla_{\boldsymbol{\theta}} f_i(\boldsymbol{\theta})\| \leq G, \forall i$, and the objective function is gradient Lipschitz continuous with constant $L$ and it is lower bounded, i.e., $f^\star = \inf_{\boldsymbol{\theta}} f(\boldsymbol{\theta}) > -\infty$.

**A. 2** We assume that gradient estimate is unbiased, i.e., $\mathbb{E}[\widehat{\nabla}_{\boldsymbol{\theta}} f_{\mathcal{B}_i}(\boldsymbol{\theta})] = \nabla_{\boldsymbol{\theta}} f(\boldsymbol{\theta})$, and has bounded variance, i.e., $\mathbb{E}[\|\widehat{\nabla}_{\boldsymbol{\theta}} f_{\mathcal{B}_i}(\boldsymbol{\theta}) - \nabla_{\boldsymbol{\theta}} f(\boldsymbol{\theta})\|^2] \leq \sigma^2$. By leveraging the contraction property of CLT-$k$, we can have following convergence rate guarantees.

**Theorem 1.** *Under assumptions A.1-A.2, suppose the sequence $\{\boldsymbol{\theta}^t\}$ is generated by CLT-k. Then, when learning rate $\alpha$ and discounting factor $\beta$ are chosen as*

$$\alpha \sim \mathcal{O}\left(\frac{\sqrt{n}}{\sigma\sqrt{T}}\right), \quad \frac{1 + \gamma - \sqrt{1 - \gamma^2}}{2(1 + \gamma)} < \beta < \frac{1 + \gamma + \sqrt{1 - \gamma^2}}{2(1 + \gamma)}, \tag{9}$$

*where $T$ denotes the total number of iterations and $0 \leq \gamma < 1$, we have*

$$\frac{1}{T} \sum_{t=1}^{T} \mathbb{E}\|\nabla_{\boldsymbol{\theta}} f(\boldsymbol{\theta}^t)\|^2 \leq \frac{\left(f(\boldsymbol{\theta}^1) - f^\star\right)\sigma}{2\sqrt{nT}} + \frac{2L\sigma}{\sqrt{nT}} + \mathcal{O}\left(\frac{1}{T}\right). \tag{10}$$

*Remark 4.* Theorem 1 showcases the *linear speedup* that can be achieved by CLT-$k$, meaning that the optimiality gap (i.e., size of gradient) is decreased as the number of workers increases. Next, we will give the analysis to show how the number of workers $n$ and corresponding correlation between the workers jointly affect the convergence in terms of $\gamma$, especially for the case when $n$ is large.

**Lemma 2.** *Let* $\mathbf{x}_i$ *denote* $\mathbf{m}_i^t + \widehat{\nabla}_{\boldsymbol{\theta}} f_{\mathcal{B}_i}(\boldsymbol{\theta})$ *and* $\mathbb{E}\|CLT_i^k(\mathbf{x}_j) - \mathbf{x}_j\|^2 \le \gamma_j \mathbb{E}\|\mathbf{x}_j\|^2, \forall \mathbf{x}_i, \mathbf{x}_j$. *Assume that gradients at different workers are positively correlated, (i.e., exists a positive constant* $\kappa$ *such that* $\mathbb{E}[\mathbf{x}_i^T \mathbf{x}_j] \ge \kappa \|\mathbf{x}_i\| \|\mathbf{x}_j\|, \forall i, j$), *and* $\mathbb{E}\|\mathbf{x}_i\|^2 = \mathbb{E}\|\mathbf{x}_j\|^2, \forall i, j$, *then if* $\kappa > (n \sum_{i=1}^n \gamma_i - 1)/(n(n-1))$ *we have* $\gamma = \frac{n \sum_{i=1}^n \gamma_i}{1 + \kappa n(n-1)} < 1$ *such that* $\mathbb{E}\|\mathbf{y} - CLT_i^k(\mathbf{y})\|^2 \le \gamma \mathbb{E}\|\mathbf{y}\|^2$, *where* $\mathbf{y} = \frac{1}{n} \sum_{i=1}^n \mathbf{x}_i$.

*Remark 5.* It can be seen that if $\sum_{i=1}^n \gamma_i \sim o(n)$ and $\kappa \sim \mathcal{O}(1)$, then $\gamma \sim \mathcal{O}(1/n)$, implying that the contraction constant is decreased w.r.t. $n$. If $\sum_{i=1}^n \gamma_i \sim \mathcal{O}(n\kappa)$, we will have $\gamma \sim \mathcal{O}(1)$, showing that in this case ScaleCom is able to find the first-order stationary point for any $\kappa > 0$.

**Discussion:** Given a pre-defined $k$, the contraction coefficient of CLT-$k$ given in (7) depends on the top-$k$ contraction coefficient $\gamma_0$ and the Hamming distance $d$. The top-$k$ contraction property has been widely investigated in literature. Theoretically, the upper bound of top-$k$ contraction $\gamma_0$ is $1 - d/n$, which is the same as random-$k$ when the components of gradient are uniform. Practically, $\gamma_0$ is observed to be a much smaller value [33].

On the other hand, the Hamming distance $d$ measures the overlap between two top-$k$ index sets. Figure 3 shows the normalized Hamming distance $d/k$ over iterations and various number of workers. The smaller the $d/k$, the closer the $\gamma$ to $\gamma_0$. It demonstrates that empirically the overlap between local top-$k$ indices from one worker and the global true top-$k$ indices after all-reduce is reasonable ($d/k$ is in the range of 0.6-0.8), which indicates a good contraction property of the CLT-$k$ compressor in practice. This will further affect the discounting factor $\beta$ in low-pass filtering as shown in Theorem 1.

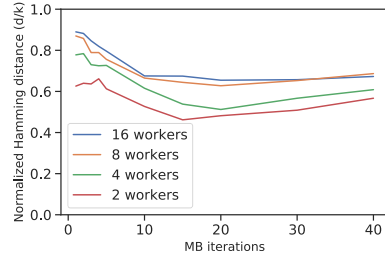

Figure 3: Normalized hamming distance between true top-$k$ and CLT-$k$, which is observed to be between 0.6-0.8. This is measured using ResNet18 on CIFAR10 with learning rate 0.1 and compression rate=400X at epoch 0. Per-worker batch size is 32.

**Large datasets and small batch size:** Large dataset/small batch size introduces more noise in gradients decreasing statistical similarity between workers and is thus harder to deal with. In the analysis above, we've assumed that the minimum overlap of Hamming distance between workers to guarantee contraction < 1, which is a mild assumption in practice. Figure 3 shows that when the per-worker batch size is 32, the Hamming distance is still above 0.32 - which is consistent with our pilot experiments, where we tried a minibatch per-worker of 8 with 128 workers on CIFAR10 without any noticeable degradation. This indicates ScaleCom's applicability in challenging training conditions (large datasets/small mini-batch size).

# 4 Experimental Results

We apply ScaleCom to three major applications: vision (ImageNet, CIFAR10), language (WMT14 En-De), and speech (SWB300). Experiments are run on IBM POWER System AC922 systems using implementations in PyTorch.[6] We adopt [39] to accelerate sorting, which divides the whole buffer into chunks and parallelizes sorting in each chunk. As suggested in [22][23], we use 1-5 warm-up epochs (<10% total training epochs) for compression.[7] A conservative engineering guidance is proposed for compression rate settings in each layer based upon the ratio *FLOPs/gradient*: 25X for ratio in the range $[196, \infty]$; 50X for $[128, 196]$, and 400X for $(0, 128)$. It should be noted that this guidance is based on the per-worker mini-batch size, 32 for vision and speech and 4.5k for language. As per-worker mini-batch size changes, the compression rate is adjusted accordingly. In addition, to demonstrate the robustness of ScaleCom, a much more aggressive compression rate for Transformer-based language model is tested in both standard and large batch size.

**Standard Batch Size:** In these experiments, we adopt hyper-parameter settings from [1][3][5] (including learning rates and momentum) to achieve excellent baseline accuracy (listed in Table 2). The same hyper-parameters are used in ScaleCom experiments, in which we set $\beta$=1 in the low-pass filter, as there is no need to filter the gradients in standard batch size experiments. The experimental results are summarized in Table 2, and convergence curves are shown in Figure 4. With compression rates of 65-400X, ScaleCom achieves accuracies very close to the baseline for all workloads.

Table 2: Baseline vs. compression standard batch size training on image, language and speech models

| Model (Dataset) Accuracy or [other metrics] | #GPU | BSZ | Comp. Rate | Baseline | Comp. |
|---|---|---|---|---|---|
| ResNet34 (CIFAR10) | 4 | 128 | 92X | 93.78 | 93.98 |
| ResNet18 (ImageNet) | 8 | 256 | 112X | 70.482 | 70.172 |
| ResNet50 (ImageNet) | 8 | 256 | 96X | 76.442 | 75.988 |
| MobileNetV2 (ImageNet) | 8 | 256 | 155X | 71.644 | 71.524 |
| Transformer-base (WMT14 En-De) [BLEU] | 8 | 36K | 47X (65X*) | 27.64 | 27.27 (27.24*) |
| 4-bidirectional-LSTM Speech (SWB300) [WER] | 4 | 128 | 400X | 10.4 | 10.1 |

*More aggressive compression is applied without significant degradation.

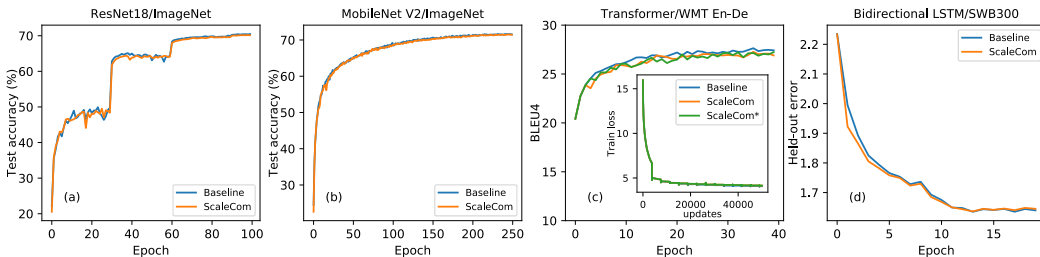

Figure 4: Standard batch size training curves with ScaleCom on (a) ResNet18 for ImageNet dataset (b) MobileNetV2 with width-multiplier 1.0 on ImageNet (c) Transformer-base machine translation (ScaleCom* corresponds to 65X in Table 2) (d) LSTM-based speech model for the SWB300 dataset. Convergence and accuracy are preserved across various models and datasets. Final training results are summarized Table 2.

**Large Batch Size Scaling:** To evaluate the scalability of our methods, we follow [7][11][40] to achieve state-of-the-art baseline accuracy with large-scale distributed settings (listed in Table 3). Compression experiments use the same hyper-parameters as baselines. From Section 2.2, as we scale up the mini-batch size and learning rates in large-scale distributed training, the gradient noise increases and local memory similarity becomes weaker among workers, which could damage network performance. As shown in the gray lines of Figure 5, when the low-pass filter is not applied ($\beta$=1), although small dataset (CIFAR10) still shows good accuracy, large datasets (ImageNet, WMT14, and SWB300) start to show degradation. Once the proposed low-pass filter is applied ($\beta$=0.1), ScaleCom achieves almost identical test accuracies when compared to the non-compressed baseline on every large network studied as shown in Table 3 and Figure 5.[8]

Table 3: Baseline vs. compression large batch size training on image, language, and speech models

| Model (Dataset) Accuracy or [other metrics] | #GPU | BSZ | Comp. Rate | Baseline | Comp. |
|---|---|---|---|---|---|
| ResNet34 (CIFAR10) | 32 | 1024 | 92X | 93.75 | 93.36 |
| ResNet18 (ImageNet) | 64 | 2048 | 112X | 70.285 | 69.879 |
| ResNet50 (ImageNet) | 64 | 2048 | 96X | 76.473 | 75.895 |
| MobileNetV2 (ImageNet) | 64 | 2048 | 155X | 71.487 | 71.014 |
| Transformer-base (WMT14 En-De) [BLEU] | 64 | 288K | 47X (115X*) | 27.79 | 28.03 (27.59*) |
| 4-bidirectional-LSTM Speech (SWB300) [WER] | 12 | 1536 | 100X | 9.9 | 10.0 |

*More aggressive compression is applied without significant degradation.

# 5 End-to-end System Performance

In this section, we quantify the improvement in end-to-end training time achieved by ScaleCom. We considered a distributed training system comprised of multiple accelerator chips connected to a parameter server. Each accelerator chip consists of multiple cores with private scratchpad memory. The systematic performance analysis framework presented in [35] is used to estimate performance. Given a system configuration (compute throughput, memory capacity, interconnect topology and

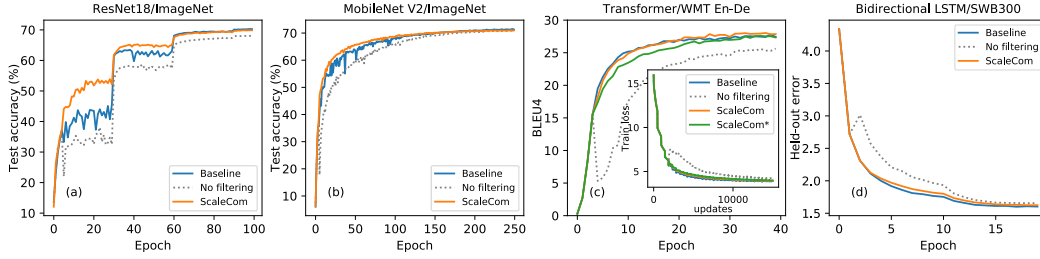

Figure 5: Large batch size training curves with ScaleCom on (a) ResNet18 for ImageNet dataset (b) MobileNetV2 with width-multiplier 1.0 on ImageNet (c) Transformer-base machine translation (ScaleCom* corresponds to 115X in Table 3) (d) LSTM-based speech model for SWB300 dataset.

bandwidth), the framework analytically explores possible ways to map DNN computations on to the accelerator system and provide performance estimations.[9]

We present the performance impact of ScaleCom by varying 3 key factors: (i) peak compute capability per worker (100 and 300 TFLOPs) (ii) the size of mini-batch per worker (8 and 32), and (iii) the number of workers (8, 32 and 128). When the mini-batch per worker is increased, the gradient/weight communication becomes less frequent, limiting the scope of end-to-end performance benefits from ScaleCom. This is evident from Figure 6a, where the communication time (as a fraction of total time) decreases from 56% to 20%, when the mini-batch per worker is increased from 8 to 32. Consequently, with 100 TFLOPs peak compute per worker, ScaleCom achieves total training speedup of $2\times$ to $1.23\times$ even with $\sim100\times$ compression ratio. Fraction of communication time grows with increase in peak TFLOPs (100 to 300), resulting in speedup of $4.1\times$ to $1.75\times$.

The key trait of ScaleCom is its *performance scalability to larger number of workers* independent of minibatch per worker. This is shown in Figure 6b, where the communication cost of prior top-k approaches increase linearly with number of workers, whereas that of ScaleCom remains constant. With Scalecom, the gradient/weight communication is < 3% of total training time even with large number of workers (128) and small mini-batch per worker (8), leaving the training throughput limited only by the computation inefficiency.

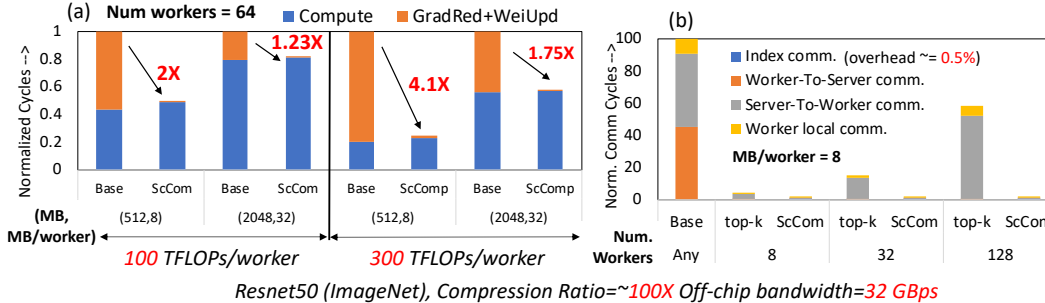

Figure 6: Stacked bar chart for Resnet50 (ImageNet dataset): (a) different per worker mini-batch sizes and (b) different worker numbers.

**Cost of index communication and synchronization:** To enable all workers to select the same gradients, ScaleCom incurs an additional overhead for communicating the top-k indices. As the index vector has the same degree of compression as the gradient vector, it occupies only 0.5% of baseline communication time. Also, the cost remains constant ($\mathcal{O}(1)$) independent of the number of workers. ScaleCom also incurs an additional synchronization during the index communication. Similar to fully synchronous SGD the slowest worker determines when the gradient communication can begin. Once this point is reached by all workers, the additional synchronization costs little extra time.

## 6 Conclusion

Gradient compression is a promising technique to resolve communication bottlenecks, but has not been widely adopted in today's training systems. The two primary reasons for this include lack of demonstrations on large batch sizes (and datasets) and the incompatibility of compression techniques with all-reduce schemes. In this paper, we propose a new compression algorithm, ScaleCom, that resolves both of these issues. We theoretically analyze ScaleCom needs as well demonstrate scalability, robustness and excellent compression rates (65-400X) using experiments on a spectrum of models, datasets and batch-sizes - laying the foundation for its introduction in large scale systems.

## Broader Impact

The amount of compute for DNNs training doubles every 3 to 4 months [41]; this is faster than Moore's law that doubles the number of transistors every 2 years. The latest language model GPT3 [42] takes 175 billion parameters to achieve state of the art performance on several NLP tasks such as common sense reasoning and word prediction. Training, designing, and optimizing these gigantic models require tremendous time (cost) and computation power. Our research results on compression in large-scale distributed training have two broad benefits:

(i) **Reducing time and cost to train DNN models:** We believe that communication times will bottleneck training times of distributed systems and this will become even more severe with recent significant improvements in the computational capability of deep learning training hardware. To address this bottleneck, in the past few years, compression techniques have been eagerly researched and implemented in some practical training systems [43]. Our research results on scalability of gradient compression aim to push this to larger scale distributed training systems, which is needed for the training of expensive and powerful gigantic models. We believe that the scalable compression solution can accelerate machine learning research and save the cost for company and research institutes to develop state-of-art DNNs in real applications and complicated datasets.

(ii) **Energy consumption for environment concerns:** Training DNNs especially for big models consumes tremendous energy and starts to cause concerns in $CO_2$ emission. As indicated in [44], Transformer training with neural architecture search could cause $CO_2$ emission as much as 5 cars' lifetime. Today most DNNs training runs in distributed systems and energy is mainly consumed in data communication: 32-bit I/O communication took 3-4 orders of more energy (pJ) than 32-bit float ADD computation [45]. Thus, efficient communication is crucial to reduce energy consumption and mitigate concerns in carbon footprint of DNNs training, especially for large-scale distributed training of gigantic DNNs. Our research cuts communication data size by 65-400X and scale this method to larger scale distribution, which will reduce energy consumption and mitigate environment concerns in gigantic DNNs training. This helps to fight climate change and global warming.

Meanwhile, we would like to point out, although our compression scheme guarantees theoretical convergence and shows no accuracy loss compared to baseline training over the tested models and applications, there could still be concerns about the impact of lossy gradient compression on neural network convergence performance. Especially when gradient compression is applied directly without fine tuning hyper-parameters, training could still be subject to instability, and thus it is recommended to examine the compression scheme over a wider range of models and applications. Our conservative compression selection rules (described in section 4) help mitigate this concern, however task-specific robustness studies are recommended for special applications.

## Acknowledgments

The authors would like to thank Jintao Zhang and Ashish Ranjan for helpful technical discussions, Kim-Khanh Tran, Anthony Giordano, I-Hsin Chung, Ming-Hung Chen, Kaoutar El maghraoui, and Jeffrey Burns for the computing infrastructure, and Leland Chang, Arvind Kumar, Yulong Li, Shubham Jain, Sunil Shukla, Ankur Agrawal, Marcel Schaal, Mauricio Serrano, Wei Wang and the team for the chip platform targeted in this work. This research is realized by generous collaborations across IBM Research. Funding of this work is fully supported by IBM Research.

## Footnotes

[1]The cosine distance of two real-valued vectors $\mathbf{x}, \mathbf{y} \in \mathbb{R}^p$ is defined as $1 - \frac{x^T y}{\|\mathbf{x}\|_2 \|\mathbf{y}\|_2}$.

[2]Sum of local memory and new computed gradients

[3]Top-2% is used here.

[4]$t$ denotes the index of iterations.

[5]Check Appendix-C for proof and Appendix-D for details in convergence analysis and theory exposition.

[6]See Appendix-E for experimental details.

[7]No compression is applied during the warm-up period.

[8]We observed that $\beta$ is robust to different networks' convergence in the range of 0.1-0.3.

[9]Appendix-F provides further details on end-to-end system performance.

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
