[Supplementary Material]

# A  Observations in Local Memory Similarity

We observed local memory's similarity through Q–Q (quantile-quantile) plots as shown in Figure A1(a)-(c). In Figure A1(a), the linearity of the points in Q-Q plot suggests that the worker 1's local memory (accumulated gradient) magnitudes have very similar statistical distributions as worker 2. The red line is the linear regression fitting for the blue dots; overall its $R^2$ score is 0.99; indicates their local memory magnitude distributions (worker 1 and 2) are very similar. This is consistent to our observations in pairwise cosine distance shown in Figure 2(a). The memory accumulation (accumulate gradients over iterations) reduces gradient variation and enhances the similarity across workers. On the other hand, when plotting the computed gradients (right after backward computation) in Q-Q plot, we do not observe the excellent similarity between worker 1 and 2 as shown in Figure A1(b). In this case, the linear regression fitting $R^2$ score is 0.89. In Figure A1(c), we compare worker 1's error-feedback gradients (local memory + computed gradients) magnitudes with global all-reduced error-feedback gradient magnitudes. From the plot, we can observe that their distribution quantiles are highly correlated ($R^2$ linear regression fitting score is 0.99). The Spearman's rank correlation coefficient between worker 1 and all-reduced error-feedback gradient magnitudes is 0.657 (p-value=0). This indicates that we can possibly use local worker's top-$k$ to approximate true top-$k$ selections.

Figure A1: Q-Q plots for local memory and gradient magnitudes on ResNet18 (CIFAR10) with 8 workers at iteration 100 in epoch 0. Local top-$k$ [1] (top 0.1%) and learning rate 0.01 are used in the experiments. Red lines are the linear regression fitting results. (a) worker 1's local memory magnitudes quantile versus worker 2; (b) worker 1's computed gradient magnitudes versus worker 2. (c) worker 1's error-feedback gradient (local memory + computed gradients) magnitudes versus all-reduced (global) error-feedback gradient magnitudes.

# B  Preliminaries

Before showing the convergence proofs, we give the following table to highlight the notations and definitions of the variables used in the proofs.

Recall the optimization problem is

$$\min_{\boldsymbol{\theta} \in \mathbb{R}^p} f(\boldsymbol{\theta}) := \frac{1}{n} \sum_{i=1}^{n} f_i(\boldsymbol{\theta}), \tag{A1}$$

and the gradient estimate is as the following

$$\widehat{\nabla}_{\boldsymbol{\theta}} f_{\mathcal{B}_i}(\boldsymbol{\theta}) = \frac{1}{|\mathcal{B}_i|} \sum_{j \in \mathcal{B}_i} \nabla_{\boldsymbol{\theta}} f_{i,j}(\boldsymbol{\theta}). \tag{A2}$$

We also use several standard inequalities as follows:

1, Young's inequality with parameter $\epsilon$ is

$$\langle \mathbf{x}, \mathbf{y} \rangle \leq \frac{\epsilon}{2} \|\mathbf{x}\|^2 + \frac{1}{2\epsilon} \|\mathbf{y}\|^2 \tag{A3}$$

where $\mathbf{x}, \mathbf{y} \in \mathbb{R}^p$ are vectors.

One variant of Young's inequality is

$$\|\mathbf{x} + \mathbf{y}\|^2 \leq (1 + \epsilon) \|\mathbf{x}\|^2 + \left(1 + \frac{1}{\epsilon}\right) \|\mathbf{y}\|^2 \tag{A4}$$

Table A1: Definition of parameters used in the proofs

| Parameter | Expression/Definition | Representation |
|---|---|---|
| $p$ | $\mathbb{R}$ | problem dimension |
| $d$ | (6) | Hamming distance |
| $n$ | $\mathbb{R}$ | total number of workers |
| $\alpha$ | $\mathbb{R}$ | step size |
| $\beta$ | $\mathbb{R}$ | discounting factor of low-pass filter |
| $\gamma$ | $\mathbb{R}$ | contraction parameter |
| $\boldsymbol{\theta}$ | $\mathbb{R}^p$ | optimization variable |
| $f_i(\cdot)$ | $\mathbb{E}_{\xi_i \sim \mathcal{D}} F(\cdot, \xi_i)$ | individual function |
| $f(\cdot)$ | (A1) | objective function |
| $\mathcal{I}^k(\cdot)$ | (2) | index set of top entries $k$ |
| $\mathcal{B}_i$ | n/a | mini-batch set at worker $i$ |
| $\widehat{\nabla} f_{\mathcal{B}_i}(\mathbf{x})$ or $\widehat{\nabla} f_i(\mathbf{x})$ | (A2) | gradient estimate by $\mathcal{B}_i$ |
| $\mathbf{m}_i$ | $\mathbb{R}^p$ | memory at the $i$th node |
| $t$ | $\mathbb{R}$ | index of iteration |
| $T$ | $\mathbb{R}$ | total number of iterations |
| $G$ | A.1 | upper bound of gradients' size |
| $L$ | A.1 | gradient Lipschitz constant |
| $f^\star$ | A.1 | global minimum of $f(\mathbf{x})$ |

2, The quadrilateral identity is

$$\langle \mathbf{x}, \mathbf{y} \rangle = \frac{1}{2} \left( \|\mathbf{x}\|^2 + \|\mathbf{y}\|^2 - \|\mathbf{x} - \mathbf{y}\|^2 \right). \tag{A5}$$

## C Proof of Lemma 1 on Contraction Property

*Proof.* Suppose $\mathbf{y}$ is an $p$-dimensional vector and $\mathcal{I}^k$ is its top-$k$ index set. $\mathbf{y}$ is sparsified by the compressor $\text{comp}(\cdot)$ via another index set $\tilde{\mathcal{I}}^k$ and $\mathbf{H}(\mathcal{I}^k, \tilde{\mathcal{I}}^k) = 2d$.

We have

$$\|\mathbf{y} - \text{comp}(\mathbf{y})\|^2 \tag{A6}$$

$$= \|\mathbf{y}\|^2 - \sum_{i=1}^{p} \mathbf{y}_i^2 I\{i \in \tilde{\mathcal{I}}^k\} \tag{A7}$$

$$= \|\mathbf{y}\|^2 - \sum_{i \in \mathcal{I}^k} \mathbf{y}_i^2 I\{i \in \tilde{\mathcal{I}}^k\} - \sum_{i \in \backslash \mathcal{I}^k} \mathbf{y}_i^2 I\{i \in \tilde{\mathcal{I}}^k\} \tag{A8}$$

$$\leq \|\mathbf{y}\|^2 - \sum_{i \in \mathcal{I}^k} \mathbf{y}_i^2 I\{i \in \tilde{\mathcal{I}}^k\}. \tag{A9}$$

Since $\mathbf{H}(\mathcal{I}^k, \tilde{\mathcal{I}}^k) = 2d$, there is an overlap of $k-d$ indices between $\mathcal{I}^k$ and $\tilde{\mathcal{I}}^k$. Taking the expectation over the all possible combinations and permutation, we have

$$\mathbb{E} \left\| \mathbf{y} - \text{comp}(\mathbf{y}) \right\|^2 \tag{A10}$$

$$= \| \mathbf{y} \|^2 - \sum_{i \in \mathcal{I}^k} \mathbf{y}_i^2 \frac{C_{k-1}^{k-d-1}}{C_k^{k-d}} \tag{A11}$$

$$= \| \mathbf{y} \|^2 - \frac{k-d}{k} \sum_{i \in \mathcal{I}^k} \mathbf{y}_i^2 \tag{A12}$$

$$= \| \mathbf{y} \|^2 - \frac{k-d}{k} \left( \| \mathbf{y} \|^2 - \| \mathbf{y} - \text{top}_k(\mathbf{y}) \|^2 \right) \tag{A13}$$

$$\leq \| \mathbf{y} \|^2 - \frac{k-d}{k} \| \mathbf{y} \|^2 + \frac{k-d}{k} \gamma_0 \| \mathbf{y} \|^2 \tag{A14}$$

$$= \left( \frac{d}{k} + \left( 1 - \frac{d}{k} \right) \cdot \gamma_0 \right) \| \mathbf{y} \|^2, \tag{A15}$$

which completes the proof. $\qquad\square$

## D    Convergence Performance Analysis of CLT-$k$

**Lemma 3.** *Under assumptions A.1-A.2. Suppose the sequences $\{\boldsymbol{\theta}^t, \mathbf{m}_i^t, t \geq 1\}$ is generated by CLT-k. Then, when $\beta \in (\frac{1+\gamma-\sqrt{1-\gamma^2}}{2(1+\gamma)}, \frac{1+\gamma+\sqrt{1-\gamma^2}}{2(1+\gamma)})$, we have*

$$\mathbb{E}\|\mathbf{m}^t\|^2 \leq \beta^2(1+\gamma)G^2 \left( 1 + \frac{1}{C_\epsilon} \right) \left( G^2 + \frac{\sigma^2}{n} \right) \frac{\lambda}{1-\lambda} \tag{A16}$$

*where $\mathbf{m}^t = n^{-1} \sum_{i=1}^n \mathbf{m}_i^t$, and $C_\epsilon > 0$ and $0 < \lambda < 1$ are some constants.*

### D.1    Proof of Lemma 3

*Proof.* Define a "virtual" sequence $\mathbf{v}$ as the following

$$\mathbf{v}^{t+1} = \mathbf{v}^t - \alpha \underbrace{\frac{1}{n} \sum_{i=1}^n \widehat{\nabla}_{\boldsymbol{\theta}} f_{\mathcal{B}_i}(\boldsymbol{\theta}^t)}_{:=\widehat{\nabla}_{\boldsymbol{\theta}} f(\boldsymbol{\theta}^t)}. \tag{A17}$$

For simplicity of notation, we denote $\widehat{\nabla}_{\boldsymbol{\theta}} f_i(\boldsymbol{\theta})$ as $\widehat{\nabla}_{\boldsymbol{\theta}} f_{\mathcal{B}_i}(\boldsymbol{\theta})$. When $\mathbf{v}^0 = \boldsymbol{\theta}^0 = \mathbf{m}^0 = 0$, it is easy to check that

$$\boldsymbol{\theta}^t - \mathbf{v}^t = \frac{\alpha}{\beta} \underbrace{\frac{1}{n} \sum_{i=1}^n \mathbf{m}_i^t}_{:=\mathbf{m}^t}, \tag{A18}$$

so we have

$$\|\boldsymbol{\theta}^t - \mathbf{v}^t\|^2 = \frac{\alpha^2}{\beta^2} \|\mathbf{m}^t\|^2. \tag{A19}$$

From A. 2, we have

$$\mathbb{E}\|\nabla_{\boldsymbol{\theta}} f(\boldsymbol{\theta}) - \widehat{\nabla}_{\boldsymbol{\theta}} f(\boldsymbol{\theta})\|^2 \leq \frac{\sigma^2}{n} \tag{A20}$$

since $\|\nabla_{\boldsymbol{\theta}} f(\boldsymbol{\theta}) - \widehat{\nabla}_{\boldsymbol{\theta}} f(\boldsymbol{\theta})\| = \|\nabla_{\boldsymbol{\theta}} f(\boldsymbol{\theta}) - n^{-1} \sum_{i=1}^n \widehat{\nabla}_{\boldsymbol{\theta}} f_{\mathcal{B}_i}(\boldsymbol{\theta})\|$.

Then, we are able to quantify $\|n^{-1} \sum_{i=1}^{n} \mathbf{m}_i^t\|$ as follows:

$$
\|\mathbf{v}^{t+1} - \boldsymbol{\theta}^{t+1}\|
$$

$$
\stackrel{\text{(A17)}}{=} \left\| \mathbf{v}^t - \boldsymbol{\theta}^t + \alpha \frac{1}{n} \left( \sum_{i=1}^{n} \text{CLT}_{\text{mod}(t,n)}^k (\mathbf{m}_i^t + \widehat{\nabla}_{\boldsymbol{\theta}} f_i(\boldsymbol{\theta}^t)) - \sum_{i=1}^{n} \widehat{\nabla}_{\boldsymbol{\theta}} f_i(\boldsymbol{\theta}^t) \right) \right\| \tag{A21}
$$

$$
= \left\| \mathbf{v}^t - \boldsymbol{\theta}^t + \frac{\alpha}{\beta} \mathbf{m}^t - \frac{\alpha}{\beta} \mathbf{m}^t + \alpha \frac{1}{n} \left( \sum_{i=1}^{n} \text{CLT}_{\text{mod}(t,n)}^k (\mathbf{m}_i^t + \widehat{\nabla}_{\boldsymbol{\theta}} f_i(\boldsymbol{\theta}^t)) - \sum_{i=1}^{n} \widehat{\nabla}_{\boldsymbol{\theta}} f_i(\boldsymbol{\theta}^t) \right) \right\|
$$
$$
\tag{A22}
$$

$$
\stackrel{\text{(A18)}}{\leq} \alpha \left\| \frac{1}{n} \left( \sum_{i=1}^{n} \text{CLT}_{\text{mod}(t,n)}^k (\mathbf{m}_i^t + \widehat{\nabla}_{\boldsymbol{\theta}} f_i(\boldsymbol{\theta}^t)) \right) - \frac{1}{n} \left( \sum_{i=1}^{n} \widehat{\nabla}_{\boldsymbol{\theta}} f_i(\boldsymbol{\theta}^t) + \mathbf{m}_i^t \right) \right\| + \alpha \left( \frac{1}{\beta} - 1 \right) \|\mathbf{m}^t\|^2
$$

$$
\stackrel{\text{(1)}}{=} \alpha \left\| \text{CLT}_{\text{mod}(t,n)}^k \left( \frac{1}{n} \sum_{i=1}^{n} (\mathbf{m}_i^t + \widehat{\nabla}_{\boldsymbol{\theta}} f_i(\boldsymbol{\theta}^t)) \right) - \frac{1}{n} \left( \sum_{i=1}^{n} \widehat{\nabla}_{\boldsymbol{\theta}} f_i(\boldsymbol{\theta}^t) + \mathbf{m}_i^t \right) \right\| + \alpha \left( \frac{1}{\beta} - 1 \right) \|\mathbf{m}^t\|^2
$$

$$
\stackrel{(a)}{\leq} \alpha \sqrt{\gamma} \left\| \frac{1}{n} \sum_{i=1}^{n} \left( \widehat{\nabla}_{\boldsymbol{\theta}} f_i(\boldsymbol{\theta}^t) + \mathbf{m}_i^t \right) \right\| + \alpha \left( \frac{1}{\beta} - 1 \right) \|\mathbf{m}^t\|^2 \tag{A23}
$$

where in $(a)$ we apply Lemma 1.

Squaring both sides of (A23), we have

$$
\|\mathbf{v}^{t+1} - \boldsymbol{\theta}^{t+1}\|^2 \stackrel{(a)}{\leq} \left( 1 + \frac{1}{\gamma} \right) \alpha^2 \gamma \left( (1+\epsilon) \left\| \frac{1}{n} \sum_{i=1}^{n} \mathbf{m}_i^t \right\|^2 + \left( 1 + \frac{1}{\epsilon} \right) \left\| \frac{1}{n} \sum_{i=1}^{n} \widehat{\nabla}_{\boldsymbol{\theta}} f_i(\boldsymbol{\theta}^t) \right\|^2 \right)
$$

$$
+ (1+\gamma) \alpha^2 \left( \frac{1}{\beta} - 1 \right)^2 \|\mathbf{m}^t\|^2 \tag{A24}
$$

$$
\stackrel{\text{(A18)}}{=} (1+\gamma)(1+\epsilon) \beta^2 \|\mathbf{v}^t - \boldsymbol{\theta}^t\|^2 + \alpha^2(1+\gamma) \left( 1 + \frac{1}{\epsilon} \right) \left\| \frac{1}{n} \sum_{i=1}^{n} \widehat{\nabla}_{\boldsymbol{\theta}} f_i(\boldsymbol{\theta}^t) \right\|^2
$$

$$
+ \alpha^2(1+\gamma) \left( \frac{1}{\beta} - 1 \right)^2 \|\mathbf{m}^t\|^2. \tag{A25}
$$

where in $(a)$ we use Young's inequality with parameter $1/\gamma$.

Taking expectation on both sides of (A25), we have

$$
\mathbb{E}\|\mathbf{v}^{t+1} - \boldsymbol{\theta}^{t+1}\|^2
$$

$$
\leq (1+\epsilon)(1+\gamma)\beta^2 \mathbb{E}\|\mathbf{v}^t - \boldsymbol{\theta}^t\|^2 + \alpha^2 \gamma \left( 1 + \frac{1}{\epsilon} \right) \mathbb{E} \left\| \widehat{\nabla}_{\boldsymbol{\theta}} f(\boldsymbol{\theta}^t) - \nabla_{\boldsymbol{\theta}} f(\boldsymbol{\theta}^t) \right\|^2
$$

$$
+ \alpha^2(1+\gamma) \left( 1 + \frac{1}{\epsilon} \right) \mathbb{E}\|\nabla_{\boldsymbol{\theta}} f(\boldsymbol{\theta}^t)\|^2 + \alpha^2(1+\gamma) \left( \frac{1}{\beta} - 1 \right)^2 \|\mathbf{m}^t\|^2 \tag{A26}
$$

$$
\leq (1+\epsilon)(1+\gamma)\beta^2 \mathbb{E}\|\mathbf{v}^t - \boldsymbol{\theta}^t\|^2 + \alpha^2(1+\gamma) \left( 1 + \frac{1}{\epsilon} \right) \mathbb{E}\|\nabla_{\boldsymbol{\theta}} f(\boldsymbol{\theta}^t)\|^2 + \alpha^2(1+\gamma) \left( 1 + \frac{1}{\epsilon} \right) \frac{\sigma^2}{n}
$$

$$
+ \alpha^2(1+\gamma) \left( \frac{1}{\beta} - 1 \right)^2 \|\mathbf{m}^t\|^2. \tag{A27}
$$

Since the size of the gradients is bounded by $G$, we have

$$\mathbb{E}\|\mathbf{v}^{t+1} - \boldsymbol{\theta}^{t+1}\|^2 \leq (1+\epsilon)(1+\gamma)\beta^2 \mathbb{E}\|\mathbf{v}^t - \boldsymbol{\theta}^t\|^2 + \alpha^2(1+\gamma)\left(1+\frac{1}{\epsilon}\right)\left(G^2 + \frac{\sigma^2}{n}\right)$$

$$+ \alpha^2(1+\gamma)\left(\frac{1}{\beta}-1\right)^2 \|\mathbf{m}^t\|^2 \tag{A28}$$

$$\overset{\text{(A19)}}{=} (1+\epsilon)(1+\gamma)\beta^2 \mathbb{E}\|\mathbf{v}^t - \boldsymbol{\theta}^t\|^2 + \alpha^2(1+\gamma)\left(1+\frac{1}{\epsilon}\right)\left(G^2 + \frac{\sigma^2}{n}\right)$$

$$+ (1+\gamma)(\beta-1)^2 \mathbb{E}\|\mathbf{v}^t - \boldsymbol{\theta}^t\|^2. \tag{A29}$$

It is obvious that when

$$(1+\epsilon)(1+\gamma)\beta^2 + (1+\gamma)(\beta-1)^2 := \lambda < 1, \tag{A30}$$

then sequence $\mathbb{E}\|\mathbf{v}^{t+1} - \boldsymbol{\theta}^{t+1}\|^2$ will be upper bounded by a constant. Therefore, we request

$$\frac{1}{1+\gamma} > (1-\beta)^2, \tag{A31}$$

i.e., $\beta > 1 - \sqrt{1/(1+\gamma)}$ such that $1 - (1+\gamma)(\beta-1)^2 > 0$. Then, we can choose $\epsilon$ small enough, i.e.,

$$\epsilon < \underbrace{\frac{1-(1+\gamma)(\beta-1)^2}{(1+\gamma)\beta^2} - 1}_{:=C_\epsilon}, \tag{A32}$$

so that $\lambda < 1$ when $C_\epsilon > 0$.

Next, in order to get $C_\epsilon > 0$, we need

$$1 - (1+\gamma)(\beta-1)^2 > (1+\gamma)\beta^2, \tag{A33}$$

which is equivalent to

$$2(1+\gamma)\beta^2 - 2(1+\gamma)\beta + \gamma < 0. \tag{A34}$$

Therefore, when

$$0 < \frac{1+\gamma-\sqrt{1-\gamma^2}}{2(1+\gamma)} < \beta < \frac{1+\gamma+\sqrt{1-\gamma^2}}{2(1+\gamma)} < 1, \tag{A35}$$

and $\beta > 1 - \sqrt{1/(1+\gamma)}$, we have $\lambda < 1$. Note that here $\frac{1+\gamma-\sqrt{1-\gamma^2}}{2(1+\gamma)}$ is always greater than $1 - \sqrt{1/(1+\gamma)}$. The reasons are as follows: first, we know that

$$2(1+\gamma) > 2\sqrt{1-\gamma^2}(1+\gamma). \tag{A36}$$

Adding $2(1+\gamma)$ on both sides, we will get

$$4(1+\gamma) > 1 - \gamma^2 + 2\sqrt{1-\gamma^2}(1+\gamma) + 1 + 2\gamma + \gamma^2, \tag{A37}$$

which implies

$$2\sqrt{1+\gamma} > \sqrt{1-\gamma^2} + (1+\gamma). \tag{A38}$$

Dividing $2(1+\gamma)$ on both sides, we can arrive at

$$\sqrt{\frac{1}{1+\gamma}} - \frac{\sqrt{1-\gamma^2}}{2(1+\gamma)} > \frac{1}{2}, \tag{A39}$$

which is the desired result.

Then, iterate (A29) gives

$$\mathbb{E}\|\mathbf{v}^{t+1} - \boldsymbol{\theta}^{t+1}\|^2 \overset{(a)}{\leq} \alpha^2(1+\gamma)\left(1+\frac{1}{\epsilon}\right)\left(G^2 + \frac{\sigma^2}{n}\right)\sum_{\tau=1}^t \lambda^\tau \tag{A40}$$

$$\overset{(b)}{\leq} \alpha^2(1+\gamma)G^2\left(1+\frac{1}{\epsilon}\right)\left(G^2 + \frac{\sigma^2}{n}\right)\frac{\lambda}{1-\lambda} \tag{A41}$$

where $(a)$ holds since $\mathbf{v}^0 = \boldsymbol{\theta}^0$, and $(b)$ is true because of $\lambda < 1$. $\qquad\square$

### D.2 Proof of Theorem 1

*Proof.* By the gradient Lipschitz continuity of the objective function, we have

$$f(\mathbf{v}^{t+1}) \leq f(\mathbf{v}^t) + \left\langle \nabla f(\mathbf{v}^t), \mathbf{v}^{t+1} - \mathbf{v}^t \right\rangle + \frac{L}{2} \left\| \mathbf{v}^{t+1} - \mathbf{v}^t \right\|^2 \tag{A42}$$

$$\overset{(A17)}{\leq} f(\mathbf{v}^t) - \alpha \left\langle \nabla f(\mathbf{v}^t), \widehat{\nabla} f(\boldsymbol{\theta}^t) \right\rangle + \frac{\alpha^2 L}{2} \| \widehat{\nabla} f(\boldsymbol{\theta}^t) \|^2 \tag{A43}$$

$$\leq f(\mathbf{v}^t) - \alpha \left\langle \nabla f(\mathbf{v}^t), \widehat{\nabla} f(\boldsymbol{\theta}^t) \right\rangle + \alpha^2 L \| \widehat{\nabla} f(\boldsymbol{\theta}^t) - \nabla f(\boldsymbol{\theta}^t) \|^2 + \alpha^2 L \| \nabla f(\boldsymbol{\theta}^t) \|^2. \tag{A44}$$

Taking expectation on both sides of (A44), we have

$$\mathbb{E}[f(\mathbf{v}^{t+1})] \overset{(a)}{\leq} \mathbb{E}[f(\mathbf{v}^t)] - \alpha \left\langle \nabla_{\boldsymbol{\theta}} f(\mathbf{v}^t), \nabla_{\boldsymbol{\theta}} f(\boldsymbol{\theta}^t) \right\rangle + \frac{\alpha^2 L^2}{2} \mathbb{E} \| \widehat{\nabla}_{\boldsymbol{\theta}} f(\boldsymbol{\theta}^t) \|^2 \tag{A45}$$

$$\overset{(b)}{=} \mathbb{E}[f(\mathbf{v}^t)] - \frac{\alpha}{2} \left( \| \nabla_{\boldsymbol{\theta}} f(\mathbf{v}^t) \|^2 + \| \nabla_{\boldsymbol{\theta}} f(\boldsymbol{\theta}^t) \|^2 - \| \nabla_{\boldsymbol{\theta}} f(\mathbf{v}^t) - \nabla_{\boldsymbol{\theta}} f(\boldsymbol{\theta}^t) \|^2 \right)$$

$$+ \alpha^2 L \mathbb{E} \| \nabla_{\boldsymbol{\theta}} f(\boldsymbol{\theta}^t) \|^2 + \frac{\alpha^2 L \sigma^2}{n}$$

$$\overset{(c)}{\leq} \mathbb{E} f(\mathbf{v}^t) - \frac{\alpha}{2} \mathbb{E} \| \nabla_{\boldsymbol{\theta}} f(\mathbf{v}^t) \|^2 - \left( \alpha - \alpha^2 L \right) \mathbb{E} \| \nabla_{\boldsymbol{\theta}} f(\boldsymbol{\theta}^t) \|^2$$

$$+ \frac{\alpha L^2}{2} \mathbb{E} \| \mathbf{v}^t - \boldsymbol{\theta}^t \|^2 + \frac{\alpha^2 L \sigma^2}{n} \tag{A46}$$

where $(a)$ is true due to the unbiasedness of the gradient estimate, in $(b)$ we use the quadrilateral identity, in $(c)$ we use the gradient Lipschitz continuity.

Combining (A41) and (A46), we have

$$\mathbb{E}[f(\mathbf{v}^{t+1})] \leq \mathbb{E}[f(\mathbf{v}^t)] - \left( \alpha - \alpha^2 L \right) \mathbb{E} \| \nabla_{\boldsymbol{\theta}} f(\boldsymbol{\theta}^t) \|^2 - \frac{\alpha}{2} \mathbb{E} \| \nabla_{\boldsymbol{\theta}} f(\mathbf{v}^t) \|^2$$

$$+ \frac{\alpha^3}{2} (1 + \gamma^2) L^2 G^2 \left( 1 + \frac{1}{\epsilon} \right) \left( G^2 + \frac{\sigma^2}{n} \right) \frac{\lambda}{1 - \lambda} + \frac{\alpha^2 L \sigma^2}{n}. \tag{A47}$$

When $\alpha \leq 1/L$, we have

$$\frac{\alpha}{2} \mathbb{E} \| \nabla_{\boldsymbol{\theta}} f(\mathbf{v}^t) \|^2 \leq \mathbb{E}[f(\mathbf{v}^t)] - \mathbb{E}[f(\mathbf{v}^{t+1})] + \frac{\alpha^3}{2} (1 + \gamma) L^2 G^2 \left( 1 + \frac{1}{\epsilon} \right) \left( G^2 + \frac{\sigma^2}{n} \right) \frac{\lambda}{1 - \lambda} + \frac{\alpha^2 L \sigma^2}{n}. \tag{A48}$$

Applying the telescoping sum over iterations $t = 1, \ldots, T$, we can get

$$\frac{1}{T} \sum_{t=1}^{T} \alpha \mathbb{E} \| \nabla_{\boldsymbol{\theta}} f(\mathbf{v}^t) \|^2 \leq \frac{f(\boldsymbol{\theta}^1) - f^\star}{2T} + \alpha^3 (1 + \gamma) L^2 G^2 \left( 1 + \frac{1}{\epsilon} \right) \left( G^2 + \frac{\sigma^2}{n} \right) \frac{\lambda}{1 - \lambda} + \frac{2\alpha^2 L \sigma^2}{n}, \tag{A49}$$

so we have

$$\frac{1}{T} \sum_{t=1}^{T} \mathbb{E} \| \nabla_{\boldsymbol{\theta}} f(\mathbf{v}^t) \|^2 \leq \frac{f(\boldsymbol{\theta}^1) - f^\star}{2T\alpha} + \alpha^2 (1 + \gamma) L^2 G^2 \left( 1 + \frac{1}{\epsilon} \right) \left( G^2 + \frac{\sigma^2}{n} \right) \frac{\lambda}{1 - \lambda} + \frac{2\alpha L \sigma^2}{n}. \tag{A50}$$

Since $\| \nabla_{\boldsymbol{\theta}} f(\boldsymbol{\theta}^t) - \nabla_{\boldsymbol{\theta}} f(\mathbf{v}^t) \| \leq L \| \boldsymbol{\theta}^t - \mathbf{v}^t \|$ and the upper bound of $\| \boldsymbol{\theta}^t - \mathbf{v}^t \|$ shown in (A41), we have

$$\frac{1}{T} \sum_{t=1}^{T} \mathbb{E} \| \nabla_{\boldsymbol{\theta}} f(\boldsymbol{\theta}^t) \|^2 \leq \frac{f(\boldsymbol{\theta}^1) - f^\star}{2T\alpha} + 3\alpha^2 (1 + \gamma) L^2 G^2 \left( 1 + \frac{1}{\epsilon} \right) \left( G^2 + \frac{\sigma^2}{n} \right) \frac{\lambda}{1 - \lambda} + \frac{2\alpha L \sigma^2}{n}. \tag{A51}$$

Let $\alpha := \frac{\sqrt{n}}{\sigma\sqrt{T}}$. We can get

$$\frac{1}{T}\sum_{t=1}^{T}\mathbb{E}\|\nabla_{\boldsymbol{\theta}}f(\boldsymbol{\theta}^t)\|^2 \leq \frac{\left(f(\boldsymbol{\theta}^1)-f^\star\right)\sigma}{2\sqrt{nT}} + \frac{3n^2}{\sigma^2 T}(1+\gamma)L^2G^2\left(1+\frac{1}{\epsilon}\right)\left(G^2+\frac{\sigma^2}{n}\right)\frac{\lambda}{1-\lambda}$$
$$+ \sqrt{\frac{1}{nT}}2L\sigma. \tag{A52}$$

When $T$ is large, $1/T$ decreases faster then $\sqrt{1/T}$, so we have

$$\frac{1}{T}\sum_{t=1}^{T}\mathbb{E}\|\nabla_{\boldsymbol{\theta}}f(\boldsymbol{\theta}^t)\|^2 \leq \frac{\left(f(\boldsymbol{\theta}^1)-f^\star\right)\sigma}{2\sqrt{nT}} + \frac{2L\sigma}{\sqrt{nT}} + \mathcal{O}\left(\frac{1}{T}\right). \tag{A53}$$

which shows a linear speed-up w.r.t. $n$. $\qquad\square$

### D.3 Proof of Lemma 2

*Proof.* Let $\mathbf{x}_j$ denote $\mathbf{m}_j^t + \widehat{\nabla}_{\boldsymbol{\theta}}f_{\mathcal{B}_j}(\boldsymbol{\theta})$ for simplicity in this proof. Then, by leveraging the linear property of CLT-$k$ operator, we have

$$\left\|\text{CLT}_i^k\left(\frac{1}{n}\sum_{j=1}^{n}\mathbf{x}_j\right) - \frac{1}{n}\sum_{j=1}^{n}\mathbf{x}_j\right\|^2$$
$$= \left\|\frac{1}{n}\sum_{j=1}^{n}\text{CLT}_i^k(\mathbf{x}_j) - \frac{1}{n}\sum_{j=1}^{n}\mathbf{x}_j\right\|^2 \overset{(a)}{\leq} \sum_{j=1}^{n}\frac{\gamma_j}{n}\|\mathbf{x}_j\|^2, \forall i, j. \tag{A54}$$

where $(a)$ is true since $\mathbb{E}\|\text{CLT}_i^k(\mathbf{x}_j) - \mathbf{x}_j\|^2 \leq \gamma_j\mathbb{E}\|\mathbf{x}_j\|^2, \forall \mathbf{x}_i, \mathbf{x}_j$

On the other side, we can expand $\|n^{-1}\sum_{j=1}^{n}\mathbf{x}_j\|^2$ directly by

$$\left\|\frac{1}{n}\sum_{j=1}^{n}\mathbf{x}_j\right\|^2 = \frac{1}{n^2}\left(\sum_{j=1}^{n}\|\mathbf{x}_j\|^2 + 2\sum_{i=1}^{n}\sum_{j\neq i}^{n}\mathbf{x}_i^T\mathbf{x}_j\right). \tag{A55}$$

Next, we try to have an upper of $n^{-1}\sum_{j=1}^{n}\gamma_j\|\mathbf{x}_j\|^2$ by $\|n^{-1}\sum_{j=1}^{n}\mathbf{x}_j\|^2$.

When $\mathbf{x}_i$ and $\mathbf{x}_j$ are positively correlated and assume

$$\frac{\mathbb{E}[\mathbf{x}_i^T\mathbf{x}_j]}{\mathbb{E}\|\mathbf{x}_i\|\mathbb{E}\|\mathbf{x}_j\|} \geq \kappa, \quad \forall i, j. \tag{A56}$$

Then, we can have

$$\mathbb{E}[\mathbf{x}_i^T\mathbf{x}_j] \geq \kappa\min\{\mathbb{E}\|\mathbf{x}_i\|^2, \mathbb{E}\|\mathbf{x}_j\|^2\}, \quad \forall i, j. \tag{A57}$$

Since all the data follow the same distribution, we assume that the sizes of the gradients at each work are very similar, i.e., $\mathbb{E}\|\mathbf{x}_i\|^2 = \mathbb{E}\|\mathbf{x}_j\|^2, \forall i, j$. Thus, we can obtain

$$\mathbb{E}\left\|\frac{1}{n}\sum_{j=1}^{n}\mathbf{x}_j\right\|^2 \geq \frac{1}{n^2}\left((1+\kappa n(n-1))\sum_{j=1}^{n}\mathbb{E}\|\mathbf{x}_j\|^2\right). \tag{A58}$$

where $n \geq 2$.

Combining (A54), we can obtain

$$\sum_{j=1}^{n}\frac{\gamma_j}{n}\mathbb{E}\|\mathbf{x}_j\|^2 \leq \underbrace{\frac{n\sum_{j=1}^{n}\gamma_j}{1+\kappa n(n-1)}}_{:=\gamma}\mathbb{E}\left\|\frac{1}{n}\sum_{j=1}^{n}\mathbf{x}_j\right\|^2. \tag{A59}$$

Therefore, when $\kappa > (n \sum_{j=1}^{n} \gamma_j - 1)/(n(n-1))$, we have

$$\mathbb{E} \left\| \text{CLT}_i^k \left( \frac{1}{n} \sum_{j=1}^{n} \mathbf{x}_j \right) - \frac{1}{n} \sum_{j=1}^{n} \mathbf{x}_j \right\|^2 \leq \gamma \mathbb{E} \left\| \frac{1}{n} \sum_{j=1}^{n} \mathbf{x}_j \right\|^2, \forall i \tag{A60}$$

so that $\gamma < 1$. $\qquad\square$

### D.4 Connections between theoretical proofs and experiments

We provided the following table to explain section 3's main results and connected them to other parts of paper. For Remark 4, linear speedup refers to that when $T$ is large enough, $1/\sqrt{nT}$ leads convergence rate. As worker number $n$ increases, required iteration $T$ linearly decreases to achieve the same convergence [14]. Our theorem 1 shows this; indicates its applicability in distributed training.

|  | Lemma1:<br>contraction property | Lemma2:<br>contraction in<br>distributed setting | Theorem1:<br>ScaleCom's convergence rate<br>same as SGD $(1/\sqrt{T})$ |
|---|---|---|---|
| **Intuition** | Higher correlation between<br>workers brings CLT-$k$<br>closer to true top-$k$ | Require positive correlation<br>between workers<br>in distr. setting | Ideally ScaleCom's noise<br>does not impact<br>final conv. results |
| **Connect to exp.** | Fig.2 and 3 show high correlation<br>so our contraction<br>is close to true top-$k$ | Fig.2 and 3 show<br>positive correlation<br>between workers | Table 1,2 (Fig4,5) verified<br>ScaleCom's convergence<br>same as baseline |

## E   Experimental Details

Our standard batch size experiments are performed on IBM servers where each node is equipped with 2 Intel Xeon E5-2640 V4 processors at 2.40 GHz, each having 10 cores, and 16 NVIDIA Tesla V100 GPUs 16 GB. The large batch size experiments are performed on IBM DCS supercomputer with 268 nodes. Each node is equipped with 2 IBM Power 9 processors clocked at 3.15 GHz. Each processor contains 20 cores with 4 hardware threads (160 logical processors per node). Within the total cluster, 252 nodes each contains 6 NVIDIA Tesla V100 GPUs 32 GB, and other nodes each contains 4 NVIDIA Tesla V100 GPUs 16 GB. All nodes are connected with EDR Infiniband.

The software platform is based upon PyTorch [2] and we implement in-house communication scheme with the compression method defined Algorithm 1. For distributed training, instead of using PyTorch DistributedDataParallel (DDP) [3], we implement our own scheme on top of OpenMPI (v3.0.2) and NVIDIA NCCL. Our communication scheme initiates process group through MPI, and uses customized *average_gradients* function defined in our communication package to compress and average the gradients (through both NVIDIA NCCL and MPI). The *average_gradients* function does the following: In each iteration, we sequentially select a leading worker in a cyclical order. The leading worker does chunk-wise quasi-sorting [4], and obtains its local top-k indices, that are broadcast to the other workers. Each worker follows the indices to compress its own local error-feedback gradients (the sum of computed gradients and local memory). Then, the selected gradients are averaged across workers through all-reduce function for model updates. Figure A2 is a demonstration of ScaleCom on MNIST dataset.

### E.1   ResNet34/CIFAR10

We adapt a PyTorch implementation of ResNet34 and the standard pre-processing of CIFAR10 dataset [5]. We use learning rate of 0.1 with decay 0.1 at epoch 82 and 122. We use non-Nesterov SGD optimizer with momentum of 0.9. For the standard batch size experiment, we use 32 per-worker batch size and 4 workers. The test accuracy at epoch 160 is 93.78% for baseline without compression. Following our compression guideline in Section 4 and with 5 epochs of compression warmup, the test accuracy achieves 93.98% at epoch 160 with compression rate of 92X. Note the first convolution layer is not compressed as it is very sensitive to compression. We keep per-worker batch size constant at 32 and increase the number of workers to 32 for CIFAR10 large batch size experiment. We linearly warm

compr: compression and communication module

```
compr.init(args.job,
           args.protocol,
           trace)
device = compr.device()
```

```
output = model(data)
loss = F.nll_loss(output, target)
loss.backward()
compr.average_gradients(model, options, resume, warmup)
optimizer.step()
```

compression options: select 1 index from chunk of 4

```
"chunk_size"        : 4,
"num_send"          : 1,
"num_sort"          : 1,
```

printouts:

```
Before average, gradients: tensor([ 0.0206,  0.0182,  0.0094, -0.0039, -0.0102,  0.0315,  0.0220,  0.0063],
       device='cuda:1')
Leading worker  selects indices:  tensor([0., 1., 1., 1., 1., 0., 1., 1.], device='cuda:1')
Before average, gradients: tensor([0.0138, 0.0161, 0.0210, 0.0238, 0.0211, 0.0233, 0.0245, 0.0242],
       device='cuda:2')
Before average, gradients: tensor([-0.0124, -0.0032, -0.0058, -0.0007,  0.0036, -0.0089, -0.0066, -0.0037],
       device='cuda:3')
Before average, gradients: tensor([0.0065, 0.0184, 0.0227, 0.0099, 0.0005, 0.0132, 0.0096, 0.0111],
       device='cuda:4')
After average, gradients: tensor([0.0071, 0.0000, 0.0000, 0.0000, 0.0000, 0.0148, 0.0000, 0.0000],
       device='cuda:3')
After average, gradients: tensor([0.0071, 0.0000, 0.0000, 0.0000, 0.0000, 0.0148, 0.0000, 0.0000],
       device='cuda:2')
After average, gradients: tensor([0.0071, 0.0000, 0.0000, 0.0000, 0.0000, 0.0148, 0.0000, 0.0000],
       device='cuda:4')
After average, gradients: tensor([0.0071, 0.0000, 0.0000, 0.0000, 0.0000, 0.0148, 0.0000, 0.0000],
       device='cuda:1')
Residual:  tensor([ 0.0000, -0.0032, -0.0058, -0.0007,  0.0036,  0.0000, -0.0066, -0.0037],
       device='cuda:3')
Residual:  tensor([0.0000, 0.0161, 0.0210, 0.0238, 0.0211, 0.0000, 0.0245, 0.0242],
       device='cuda:2')
Residual:  tensor([0.0000, 0.0184, 0.0227, 0.0099, 0.0005, 0.0000, 0.0096, 0.0111],
       device='cuda:4')
Residual:  tensor([ 0.0000,  0.0182,  0.0094, -0.0039, -0.0102,  0.0000,  0.0220,  0.0063],
       device='cuda:1')
```

Figure A2: Demonstration of ScaleCom with a simple model on MNIST dataset. The model contains two convolution layers and two linear layers. For demonstration, only the first 8 gradients of iteration 0 are printed here. In iteration 0, leading worker is 'cuda:1'. It selects index 0 of the first chunk and index 1 of the second chunk from chunk-based sorting, and sends them to the other three workers. All four workers used the selected indices of 'cuda:1' for error-feedback gradients compression.

up learning rate from 0.1 to 0.8 in the first 5 epochs, and follow the same learning rate decay rule as in the standard batch size setting. The test accuracy achieves 93.75% at epoch 160 when compression is not applied. Keeping all the hyper-parameters the same as no-compression experiment, ScaleCom achieves test accuracy of 93.36% with 10 epochs of compression warmup without applying low-pass filtering ($\beta$=0). Training curves are shown in Figure A3.

### E.2 ResNets/ImageNet

We adapt ResNets v1 including ResNet18 and ResNet50 from the models of torchvision [6] and the standard pre-processing of ImageNet ILSVRC2012 dataset [7]. We use learning rate of 0.1 with decay 0.1 at epoch 30, 60, and 90. We use non-Nesterov SGD optimizer with momentum of 0.9 and weight decay of 1e-4. In the standard batch size experiment, we use 32 per-worker batch size and 8 workers. The baseline test accuracy at epoch 100 is 70.482% and 76.442% for ResNet18 and ResNet50 respectively. With ScaleCom and 5 epochs of compression warmup, ResNet18 achieves test accuracy of 70.172% with compression rate of 112X, and ResNet50 achieves 75.988% with compression rate of 96X. Our large batch size setting keeps per-worker batch size constant at 32 and uses a total of 64 workers, achieving 2048 total batch size. We linearly warm up learning rate from 0.1 to 0.8 in the first 5 epochs of training. Without compression, ResNet18 and ResNet50 achieve test accuracy of 70.285% and 76.473% at epoch 100. Hyper-parameters for compression experiments are kept the same as the no-compression experiments. For ResNet18, with 5 epochs of compression warmup, if no filtering is applied, test accuracy is 68.121% at epoch 100, showing about 2.164% degradation. If $\beta$ is set to 0.1 for low-pass filtering, ScaleCom achieves 69.879% test accuracy. $\beta$=0.1 low-pass filtering helps improve test accuracy by 1.758% for ResNet18. For ResNet50, with 10 epochs of compression warmup, we achieve test accuracy of 75.641% when $\beta$ is set to 0.1. We

Figure A3: (a) Standard and (b) large batch size training curves with ScaleCom on ResNet34 for CIFAR10 dataset.

then further tune $\beta$ by increasing it to 1 at epoch 60, and achieves a better accuracy of 75.895%. The reason behind this is as we decay learning rate over the training process and low-pass filter the gradients, the network becomes less responsive in the later epochs. To prevent that, we increase $\beta$ to 1 at epoch 60 for ResNet50. Training curves at shown in Figure A4.

### E.3 MobileNetV2/ImageNet

We adapt a PyTorch implementation of MobileNetV2 and the standard pre-processing of ImageNet ILSVRC2012 dataset [8]. We use learning rate 0.0045 with 0.98 decay each epoch. We use RMSPROP optimizer with $\epsilon$ 1.0, momentum 0.9, and weight decay 4e-5. We use the full size model with width-multiplier 1. In the standard batch size experiment, we use 32 per-worker batch size and a total of 8 workers. The baseline test accuracy achieves 71.644% at epoch 250. Following compression guidelines and with 5 epochs of compression warmup, ScaleCom achieves test accuracy of 71.524% with compression rate of 155X. In the large batch size experiment, we scale the number of workers to 64 workers while keeping per-worker batch size at 32, achieving 2048 total batch size. We linearly warm up learning rate from 0.0045 to 0.036 in the first 5 epochs, and then follow the same decay factor of 0.98 per epoch. Keeping the other hyper-parameters the same as in the standard batch size settings, we achieve test accuracy of 71.487% at epoch 250 without applying compression. Then following the same compression guidelines and with 5 epochs of compression warmup, if no filtering is applied, test accuracy is 70.976%. With $\beta$=0.1 low-pass filtering, ScaleCom achieves 71.014%. Training curves are shown in Figure A5.

### E.4 Transformer/WMT14 En-De

We adapt the implementation of FairSeq [9] and use the Transformer Base model for the WMT14 En-De translation task. We use Adam optimizer. BLEU score is calculated at each epoch with beam 4, length penalty of 0.6, and remove bpe option after compound splitting [10]. In the standard batch size experiment, learning rate is 0.0007 and we use 500 updates for warmup. Per-worker batch size is 2250 and update frequency is 2. Baseline achieves BLEU score of 27.64 at epoch 40, and ScaleCom achieves 27.27 with compression rate 47X following our compression guidelines. To demonstrate the robustness of ScaleCom, we apply a more aggressive compression rate of 65X, and achieve BLEU score of 27.24. In the large batch size experiment, the number of workers is increased to 64 while per-worker batch size kept the same. We use 4000 updates for learning rate warm up to 0.0007. Without compression, we achieve 27.79 BLEU score at epoch 40. Following our compression guideline, if no filtering is applied, we observe degradation, getting 25.65 BLEU score. With $\beta$=0.1 filtering, we achieve 28.03 with compression rate of 47X, showing big improvement in

Figure A4: Training curves with ScaleCom for ImageNet dataset: (a) standard batch size on ResNet18 (b) large batch size on ResNet18 (c) standard batch size on ResNet50 and (d) large batch size on ResNet50.

BLEU score. A more aggressive compression is applied with 115X compression rate, and BLEU score 27.59 is achieved. Training curves are shown in Figure A6.

### E.5 LSTM/SWB300

We adapted the acoustic LSTM model of IBM speech [11] that contains 4 bi-directional LSTM layers and 2 fully-connected layers in the main network; each LSTM layer contains 1024 cells with 512 on each direction. On top of the LSTM layers, there is a linear bottleneck layer with 256 hidden units followed by a SoftMax output layer with 32K units corresponding to continuous density hidden Markov model (CD-HMM) states. We used input dimensions 140 and 260 in the standard and large batch size experiments respectively. For large batch size training, our learning rate schedule follows [12]; the learning rate starts at 0.1 and gradually reaches 0.8 (increase 0.07 in each epoch). Then learning rate starts to decrease by the ratio of $1/\sqrt{2}$ in each epoch. For standard batch size, we follow [11], which keeps learning rate as 0.1 for the first 9 epoch (annealing); then decrease $1/\sqrt{2}$ in each epoch. In standard batch size, per-worker batch size is 32 and 4 workers are used (total batch size=128). For large batch size experiments, per-worker batch size increases to 128 and 12 workers are used (total batch size=1536). LSTMs are unrolled 21 frames and trained with non-overlapping feature sub-sequences of that length. The 300-hour switchboard data set (SWB300) is used to train network. The training set consists of 262 hours of Switchboard 1 audio with transcripts. The test set is the 2.1-hour switchboard (SWB) data from 40 speakers. Based on our conservative compression ratio rules (described in section 4), 400X and 100X compression rate are selected for standard and large batch size respectively and 2 epoch warm up (no compression) are applied. Training curves are shown in Figure A7(a) and (b).

Figure A5: (a) Standard and (b) large batch size training curves with ScaleCom on MobileNetV2 for ImageNet dataset.

Figure A6: (a) Standard and (b) large batch size training curves with ScaleCom on Transformer Base model for WMT14 En-De translation task.

We evaluated word error rate (WER) at epoch 20. 4-gram language models (LMs) are used in decoding and acoustic weight is chosen as 0.05. In standard batch size, our baseline (no compression) obtains 10.4% WER on SWB300 test set and 17.9% on the CallHome (CH) test set. ScaleCom reaches 10.1% WER on SWB300 and 17.9% on CH test set. For large batch size experiments, our baseline achieves 9.9% WER on SWB300 test set and 17.6% on the CH test set; ScaleCom reaches 10.0% WER on SWB300 test set and 17.6% on the CH test set.

## F    Performance Benefits with ScaleCom

In this section, we quantify the improvement in end-to-end training time achieved by ScaleCom on large-scale deep learning systems. To this end, we considered a distributed training system comprised of multiple accelerator chips connected to a parameter server. Each accelerator chip is comprised of multiple cores with private scratchpad memory, delivering ∼100 TFLOPS FP16 peak performance. To demonstrate the scalability benefits of ScaleCom, we study systems with varying number of accelerator chips from 8 to 128 increasing the peak compute from 800 TFLOPS to 12.8 PFLOPS.

Figure A7: Training curves with ScaleCom on 4-bidirectional LSTM acoustic model for 300-hour switchboard (SWB300) dataset: (a) standard and (b) large batch size.

Further, we also analyze performance impact under two different accelerator ↔ parameter server bandwidths *viz.* 32 GBps (PCIe Gen4) and 64 GBps (PCIe Gen5).

We utilized the systematic performance analysis framework for DNN accelerators presented in [35] to estimate performance. Given a system configuration (compute throughput, memory capacity, interconnect topology and bandwidth), the framework analytically explores possible ways to map DNN computations on to the accelerator system. It systematically captures computations executed in each chip and data communicated through each interconnect link using a bandwidth-centric performance model to provide an estimate on performance. The key optimizations explored by the framework include the choice of data vs. model parallelism, data-structure placement in on-chip memory to optimize intra- and inter-layer data reuse, and software pipelining to overlap communication with compute. In addition to conventional uncompressed gradient reduction scheme, the framework models performance under 2 different gradient compression schemes *viz.* Local top-$k$ [1] (which encounters the gradient build-up problem outlined in Section 1) and ScaleCom.

## F.1 End-to-end Training Performance

Figure A8 shows the normalized performance improvement in end-to-end training time using different gradient compression schemes for the ResNet50 (ImageNet) benchmark as the number of workers in the system and the accelerator-to-parameter server bandwidth are varied. The performance is normalized to the case where no compression is employed with 8 workers and 32 GBps bandwidth. We present performance under *strong scaling i.e.*, the minibatch size is increased proportional to the number of workers keeping minibatch/worker constant (=8 in this experiment). To provide a fair performance comparison, we assume the same gradient compression ratio of 112× for both local top-$k$ and ScaleCom gradient compression schemes.

As expected the conventional uncompressed gradient reduction scheme scales quite well with the number of workers because the gradients are reduced in the parameter server and so the accelerator to parameter server communication cost remains constant. We observe about 1.35× improvement in performance when the bandwidth is increased from 32 to 64 GBps. Given the high compute throughput of the accelerator chip, the communication between the parameter server and accelerator is the key bottleneck to performance accounting for ∼55% of the total execution time. The gradient compression schemes are targeted to address this bottleneck yielding significant performance gains. Local top-$k$ scheme achieves about 1.92× performance improvement relative to the conventional scheme especially when the number of workers is small. However, with increase in the number of workers, each worker could select a different gradient index during compression, resulting in a linear increase in the amount of data communicated between the parameter server and the accelerators. This leads to the overall performance benefits due to compression dropping from 1.92× with 8 workers to

Figure A8: Speedup using different gradient compression schemes for Resnet50 DNN

$1.2\times$ with 128 workers. In contrast, ScaleCom overcomes the gradient-build up issue, by forcing all workers to select the same set of indices during compression, albeit incurring a small overhead for communicating the indices beforehand. This results in $2\times$ speedup relative to the conventional scheme, which is maintained independent of the number of workers. With 128 workers, ScaleCom achieves $1.65\times$ and $1.37\times$ performance gain over local top-$k$ at 32 and 64 GBps, respectively.

### F.2 System Performance in different mini-batch sizes and worker numbers

Appendix-F.1 shows ScaleCom's scalability in system performance; more details here for practical applicability in different per-worker mini-batch sizes and worker numbers. The fraction of time expended in gradient/weight communication limits the overall end-to-end training time improvement achieved with ScaleCom. As shown in Figure A9a, when minibatch/worker is increased from 8 to 32, the communication time (as a fraction of total time) decreases from 56% to 20%. Consequently, for a 100 TFLOPs/worker peak compute capability, ScaleCom achieves total training speedup of $2\times$ to $1.23\times$ even with $\sim100\times$ compression. Fraction of communication time grows with increase in peak TOPs (100 to 300), resulting in speedup of $4.1\times$ to $1.75\times$. The key trait of ScaleCom is its *performance scalability to larger number of workers* independent of minibatch/worker. As shown in Figure A9b, the communication cost of prior top-k approaches increase linearly with number of workers, whereas ScaleCom remains constant. ScaleCom offers scalability even with large number of workers independent of per-worker minibatch size and effectively reduces communication cycles (< 3% of total training time) leaving the training throughput to be limited by the computation efficiency.

Figure A9: Staked bar chart for Resnet50 DNN: (a) different per worker mini-batch sizes and (b) different worker numbers