[Reviews · NeurIPS 2020]

Review 1

Summary and Contributions: The paper considers distributed learning and proposed a sparsification method to reduce the communication overhead. The proposed method sparsifies the stochastic gradients of all workers based on the sparsity pattern of a selected worker. Assuming that there exists some similarity among the gradients computed across the workers, the proposed global sparsification can yield low MSE. To account for the error during compression, the authors also utilized error feedback with low-pass filtering in their compression scheme. The proposed algorithm is evaluated both theoretically and via simulations.

Strengths: Distributed training has been of interest to the ML and optimization community in the past several years. The paper proposes a new method which is compatible with all-reduce techniques. The paper is well-written and the proposed technique is motivated and explained clearly. Although the building blocks of the proposed method are well-known, the main novelty is in the cyclic top-k sparsifier which uses the sparsity pattern of a selected worker to sparsify the other workers' gradients. The theoretical results look correct and the simulations show the relatively good performance of the proposed technique compared to the baseline distributed training.

Weaknesses: 1- The algorithm is based on the assumption that the gradients and error-feedback at different workers are correlated. This may be true for large batch-sizes, but generally, for small batch-sizes and large datasets, this might not hold. Hence, at the initial stages of training, the proposed method may not perform well. 2- In addition to the mentioned references in the paper, there are several other works on weighted error feedback, such as [arXiv:1806.08054] and [doi.org/10.1609/aaai.v34i04.5706]. For example, the latter analyzes the weighted error feedback with similar assumptions, which is closely related to the low-pass filtered error feedback. By comparing the theoretical convergence results, it seems that the weighted error feedback can achieve a better convergence rate at the extra term, O(1/T), compared to O(1/\sqrt(T)) in the proposed low-pass filtering feedback. 3- There are other similar works such as global top-k sparsification [arXiv:1901.04359] and techniques based on compressive sampling and random transformation for gradient compression in distributed learning. From a theoretical point of view, it looks like that these algorithms can be easily extended to be compatible with all-reduce and perform better than the proposed method. Although a good amount of simulations are provided, but they are only compared to the baseline. It would be nice if the authors include few comparisons w.r.t. some of the other state-of-the-art compression techniques. 4- Line 3 of Algorithm 1 states that the workers do their jobs in parallel. However, to compute $CLT_j^k(.)$ at worker i, it is required that the worker j = mod(t, n) computes its gradient, finds its sparsity pattern, and then shares it with other workers. Hence, the algorithm is not fully parallelizable; it is a mix of sequential and parallel computations. ================================== I would like the authors for their responses. Regarding my concerns, I disagree with the authors on the error feedback. By simply setting m^t=\beta r_t , where r_t is as in paper [doi.org/10.1609/aaai.v34i04.5706], the same equations will be derived. Regarding motivating the LP filter to reduce SG noise, the main question that arises is that whether the SG noise is hurting the convergence or the staleness of gradients at initial stages of training where the change in parameters is relatively large. In terms of experiments, I believe that comparing with baseline is not sufficient to evaluate. Is the proposed method better than random sparsification or random transformation, where all workers share the same random number generation seed, and hence unlike the proposed method, they don't need two rounds of communications? I recommend the authors to compare their results with other well-known techniques.

Correctness: Yes

Clarity: Yes

Relation to Prior Work: To some extent. There are some key works that are missed.

Reproducibility: Yes

Additional Feedback: 1- One of the issues with the proposed method is that in the paper, there is no result to show that the residue signal $m_i^t$ are bounded. Because of the positive feedback, if the error of the compression algorithm is large, the residue signal $m_i^t$ can become unbounded. Although for the proposed compression technique, the proof might be trivial, including that can strengthen the paper. 2- In the proof of Lemma 1, it is assumed that $\tilde{I}^k$ is random with a uniform distribution over the set of possible values. It should be stated in the lemma. 3- Known results on using (weighted) error feedback shows that it can reduce the convergence gap to the baseline from $O(1/\sqrt{T})$ to $O(1/T)$. But the proposed algorithm doesn't show such a benefit. Is there any explanation for that? 4- Line 220, $\sum_i \gamma_i$ is O(n), not o(n). 5- Please state the batch-size used to generate Fig. 2 6- Line 142: the text doesn't match the equation (3) 7- Remark 3: the authors should explain more about the details. There are a few ways to extend the algorithm to a ring all-reduce. It is not clear which one will perform better.


Review 2

Summary and Contributions: The authors propose ScaleCom a framework for efficiently compressing _x0011__x0011__x0011__x0011__x0011__x0011__x0011__x0011__x0011_gradients which are communicated during distributed training, allow better scaling hence more efficient training. Specially this work aims to address the issues of gradient build-up and applicability of such gradient compression method for large batch training. The proposed algorithm is based on the following key observations - 1. during distributed training, the local gradient are computed from samples drawn from the same training set, which are sampled from sufficient stochastic to ensure similarity; 2. Based on the local gradient similarity, the local worker’s top-k indices can be used to approximate the global top-k indices, with reasonably tight bounds on accuracy; 3. Furthermore, the authors provide the insight that with larger batch training increases the gradient variance due to the commensurately larger learning rates, which potentially breaks local similarity assumption, which is the primary challenge in getting such gradient compression methods to work with lager batches. Building on the following key observation, this paper specifically - - Proposed a novel commutative *Cyclic local Top-k* compression framework for the all reduce operation of the distributed gradient synchronization - Additionally propose the use of low-pass filtering for local memory accumulation of the gradient all reduce to address the challenge of applying this method to large-batch training with With these the paper provides compelling result to show the efficacy of the proposed ScaleCom method with experiments with a wide range of large’ish applications, demonstrating upto 400X reduction in gradient traffic over the network.

Strengths: - The paper is well written, its is easy to follow - the rationale, proposed methodologies, challgenes and how they are addressed are all easy to follow. Further the authors do good job of complementing the proposed methodology with results to justify the claims - Provide a theoretical guarantee for convergence (albeit with simplifying assumptions), this is helpful and provides confidence in the proposed method - Evaluate a wide variety of application from most of the important classes of DL applications/model incl. CV (resents, MobileNet), NMT/NLP (LSTMs, Transformers) - This paper addresses a key gap in such studies which is apply the method to more practical case of large batch training, which is typically the prevalent mode of training for most of the practical applications/deployment of model training in general.

Weaknesses: - Although the authors do a good job of providing results for the proposed methodology and compare against baseline to establish the functional effectiveness. The performance comparison seems to be missing. One of the primary claims is the with the reduction of gradient volume, the scaling efficiency would improve. However, there aren’t substantial results to back this, there than the limited detail in Appendix.F. Which has comparison for a family small minibactch size per work = 8, which is almost an order of magnitude smaller than typically used configurations. Also, with the lower batch sizes the comms bottlenecks get further exposed. So the question whether the results are skewed towards showing unrealistic benefits. - The amount reduction in communication volume is quite significant, however allreduce is a portion of the total iteration time. With upto 400X compression, it would be good to see the commensurate benefit in scaling. And where the benefit taper due to the Amdhal fraction (contribution from the other components). The results from Appendix seems to indicate a max of 2X benefits, in which the regime of demising returns could be better highlighted. - Furthermore, along similar lines it would be good if there were results which showcase the ability to scale further with this comms volume reductions This is a pretty significant gap, which reduces confidence in the method due to the question around practical applicability.

Correctness: Funcationality-wise the proposed methodology seems to correct, however there are some open-questions around the performance implication (detailed in the section above)

Clarity: The paper is written well, very easy to follow and is structured nicely.

Relation to Prior Work: This paper does a good job of acknowledging and comparing most of the recent works in this area.

Reproducibility: Yes

Additional Feedback: The proposed methodology seems very promising, given that the primary benefits is coming from improved scaling, the lack of more elaborate performance studies/results is a considerable gap. It would be great if the authors could address this and provide more performance analysis and elaborate on the practical applicability of the proposed method In addition to Allreduce, given the increased use of model-parallelism prompted by the model growth, looking at the other comms patterns such as all gather, reduce-scatter would also be beneficial.


Review 3

Summary and Contributions: This paper proposes a novel gradient compression scheme, ScaleCom, to address two problems: (1) the compression ratio decreases linearly with the number of workers since each worker compresses the gradients at different indices, and (2) the noise induced by both gradient compression and scaled learning rate leads to a large accuracy degradation or even model divergence in large batch size setting. The proposed compression scheme is empirically evaluated on a wide range of large size models and datasets, including vision, speech and language tasks. The compression ratio is 65-400X. Overall, the scheme is shown to be scalable for distributed training with up to 64 workers. For the first problem, the paper proposes a cyclic local top-k compressor, CLT-k, which picks a leading worker and shares its top-k indices across all workers. The leading worker is selected in a cyclic order. The proposed solution is based on the observation that the top-k indices in local memory become very similar among different workers as the training proceeds. For the second problem, which reduces the local memory similarity, the paper applies a low pass filter to smooth the changes in gradients. It then shows that the local memory similarity is preserved after adding the filter.

Strengths: The observation on local memory similarity between workers matches the intuition that the top-k indices should be similar if data are drawn from the same distribution. The proposed solutions for sharing top-k indices and mitigating gradient noises are lightweight but effective. The compression scheme shows a constant scalability over a large number or workers. The empirical evaluations are especially strong, benchmarking the method on vision, speech, and language. As claimed in the paper, it is the first work that evaluates the gradient compression technique on large size models and datasets in practice. The theoretical analysis provides extra depth on the effectiveness of the method. The supplementary materials add review for additional mathematical tools and proofs for the lemmas included in the main body.

Weaknesses: The proposed low pass filter is actually pretty similar to momentum SGD, though the use cases are different. Perhaps there should be a discussion on this part in the paper. Edit: The authors addressed this and will consider it in a revision.

Correctness: The claims, method, empirical methodology are correct.

Clarity: Overall, the paper is well-written. The introduction clearly states the motivation for introducing ScaleCom, the technical contributions are explicitly stated, and the overall structures of the paper builds topics hierarchically in a digestible way. In particular, Figure 1 is useful in understanding the technical setup. However, the paper could briefly introduce some of the technical details about distributed training, the error feedback process, and all reduce setups to make it more accessible to readers from related fields. The equations might be overscripted at times, and perhaps some the subscripts for example could be assumed for notational clarity.

Relation to Prior Work: This work discusses previous papers on gradient compression and their limitations. It is built on top of error-feedback gradient compression.

Reproducibility: Yes

Additional Feedback: In the section on low pass filtering, the new hyperparameter beta is introduced. My major concern is on the sensitivity of different models and datasets to this beta. In Figure 2c, beta equals to 0.01 preserves the cosine similarity the best on CIFAR-10 dataset. In the large batch size experiments (Table 2), beta is set to be 0.1. How does beta exactly affect the convergence of a model? For a given model and a dataset, is it required to change beta? It is interesting that noisy gradients in large batch size setting and filtered compressed gradients have the same convergence rate. If only adding a low pass filter to the gradients without compression, would that help the model converge faster than the baseline? In Figure 4, it is a little confusing to switch from accuracy in (a) to error in (b), especially since Table 1 uses accuracy for the same data as well. In 2(i), it is mentioned that gradient residue is correlated since the local gradients are drawn from the same data distribution. Does this imply that this method should be more useful on simpler datasets, since simpler datasets should have a tighter distribution and more correlation? Edit: The authors addressed a few of these questions in their rebuttal.


Review 4

Summary and Contributions: This paper proposes a two ingredient method for straightforward distributed DNN training within an all-reduce/parameter server framework: gradients are sparsified using a cyclical top-k scheme (each worker sets the global top-k indices in turn based on its local gradient). A low-pass filter on local gradient residuals is also added. Theoretical results attempt to show constant scalability with guaranteed convergence. And empirical results on a number of datasets and networks show similar convergence to vanilla distributed learning with a roughly 2x speed up (for a fixed number of workers).

Strengths: The ideas are simple and seems to work reasonably well in practice. This would be a reasonable method for speeding up distributed training in a variety of settings without much tuning.

Weaknesses: As noted below, Section 3 is difficult to follow and there is a lack of context about how the results fit in the existing literature. (a) What is the impact of the additional sync to distribute the top-k? How might this scale with really large numbers of workers?

Correctness: () Section 3 (theory) is quite dense and lacks necessary exposition. Combined with my limited background with this sort of convergence analysis, I was not able to verify the correctness of this section. I would ask for both better text around the theoretical results to explain the intuition, as well as some explanation connecting these results to experimental results. Two specific questions: (b) I don’t see why Remark 4 (linear speed up) follows from Theorem 1 (magnitude of the gradient decreasing w/ 1/sqrt(n) for n workers) (c) How does is the main result (constant scalability) reflected in the experimental results?

Clarity: Much of the paper is well written and clear, however Section 3 (theory) is not sufficiently clear, at least to this reader who isn’t very familiar with the literature in this area

Relation to Prior Work: (d) Relation to prior work is given a brief treatment in Section 1.1 and Table 3. There isn’t a quantitative comparison regarding convergence, timing, scalability, or speedup. If I’m using another distributed method from Table 3, how should I expect my results to change when switching to ScaleCom? What is I need to use an exceptionally large number of workers? What about a small number of workers?

Reproducibility: Yes

Additional Feedback: Notes: line 6: "fail to evaluate model fidelity" is unclear line 55: "Error feedback (also referred to as 'residues')": should this be "local memory" instead of "error feedback"? Also, “residues” seems like a more intuitive term. Either way, some additional explanation of this concept and how it’s used should be explicitly stated earlier in the paper line 55: "However" is an odd transition given the logic of the paragraph Fig 1(a): the nature and/or cause of gradient buildup isn't described in the figure line 108: swap the order of "There are two advantages..." and Eq (1) for clarity line 112: "beside[s]" line 113: "contraction properties" isn't explained Fig 2: label (a), (b), ... line 166: maybe “local memory variable” instead of “memorized variable” eq (5): may be more intuitive if stated as in line 6 of Alg 1 Alg 1, line 4: minibatch Alg 1, between lines 4 and 5: Compute <gradient> Distribute CLT^k_{mod(t,n)} line 179: These need justification: “low-pass filter ... improve[s] workers’ memory similarity and smooth[s] out abrupt noise induced by scaled learning rates” eq (6): make clear that d is used by definition, not derived Fig 3, caption: omit period after “between” line 247: “FLOPS to the number of gradients” isn’t clear line 255: “... experiments[,] in which we ...” line 256: “standard batch [size] experiments” Table 3: technically this isn’t an experimental result, so it should be moved earlier in the paper, e.g. section 1 or 2

[Author Response · NeurIPS 2020]

**Due to space constraints we only address major concerns; all suggestions will be included in the final version.**

**Q1(R1) novelty of low pass (LP) filter:** The proposed LP filter is fundamentally different from previous weighted
error-feedback work [arXiv:1806.08054] and [doi.org/10.1609/aaai.v34i04.5706]. Our method aims to mitigate the impact
of increased gradient noise in large batch size training so it is necessary to apply a discounting factor $\beta$ (ex: 0.1)
in **new incoming residues**: $m_i^{t+1} = (1 - \beta)m_i^t + \beta e_t$. Previous work adds a forgetting factor $\beta$ (<1) in **error feed-**
**back** to bound the variance of **previous** residues, but it does not apply any discounting factor to incoming residues:
$m_i^{t+1} = (1 - \beta)m_i^t + e_t$. Fig. 2(c) shows clearly that applying $\beta$ to **new incoming residues** is critical for improved
local memory correlation with high learning rates. Experimentally we've observed that when using previous weighted
error-feedback, large MB ResNet18 (ImageNet) shows 3.7% degradation compared to proposed LP filter for 64 workers
(proposed LP filter: 69.8% vs previous weighted feedback: 66.1%). In theory, compared to prior arts, the extra term
resulted from the model compression is also shrinking in a rate of $\mathcal{O}(1/T)$. Please refer to eq.(A52) in appendix D.

**Q2(R1) CLT-$k$ and other top-$k$ methods:** Compared to previous top-$k$ methods (ex:[arXiv:1901.04359]), CLT-$k$ has
two major differences: (i) CLT-$k$ is a commutative operator so network convergence is guaranteed. As suggested in
eq.8 of [arxiv.org/pdf/1809.10505.pdf], without explicit assumptions, non-commutative compressors do not guarantee
convergence. (ii) CLT-$k$ has $\mathcal{O}(1)$ in both scalability and compression overhead (due to local sparsity patterns). To
approximate top elements, techniques such as gTopk and powerSGD require merging local top$k$ elements, which incurs
non-perfect scalability such as $\mathcal{O}(\log(n))$. We will compare and cite related work (gTop-$k$) in the final draft.

**Q3(R1) Remark3 all-reduce ring:** In ring all-reduce, we divide the gradient buffer into n (worker number) parts and
assign each worker a part. In the 1st iteration of reduce-scatter phase, each worker selects top-$k$ in its corresponding
piece and sends selected indexes/gradients to the next worker. Then in the following iterations of reduce-scatter, each
worker will just receive the incoming indexes/gradients, sum them with local gradients; then send results to the next one.
In each mini-batch iteration, we re-assign the piece amongst workers. Additional top-$k$ index exchange is not needed.

**Q4(R1, R3) Large datasets/small batch size:** In theory, large dataset/small batch size introduces more noise to
gradients and deceases statistical similarity between workers and is thus tougher to deal with. In sec.3 we assume min.
overlap of hamming dist. between workers to guarantee contraction < 1, which is a mild assumption in practice. Fig.3
shows that in per-worker MB=32; the hamming dist. is still above 0.32. In pilot experiments, we even tried per-worker
MB=8 on CIFAR10 without noticeable degradation. In addition, Table 1,2 had broadly reported results on large datasets
(ImageNet, WMT14). These empirical observations are consistent to [arxiv.org/pdf/1712.06559.pdf], which proved that
SGD has a small critical batch size to approximate a full gradient descent iteration, no matter the size of dataset.

**Q5(R2, R4) System performance:** Appendix-F shows ScaleCom's scalability in system performance; more
details here for practical applicability. The fraction of time expended in gradient/weight communication
limits the overall end-to-end training time improvement achieved with ScaleCom. As shown in Figure a,
when minibatch/worker is increased from 8 to
32, the communication time (as a fraction of
total time) decreases from 56% to 20%. Con-
sequently, for a 100 TFLOPs/worker peak
compute capability, ScaleCom achieves total
training speedup of 2× to 1.23× even with
~100× compression. Fraction of communi-
cation time grows with increase in peak TOPs

*Resnet50 (ImageNet), Compression Ratio=~100X Off-chip bandwidth=32 Gbps*

(100 to 300), resulting in speedup of 4.1× to 1.75×. The key trait of ScaleCom is its *performance scalability to larger*
*number of workers* independent of minibatch/worker. As shown in Figure b, the communication cost of prior top-k
approaches increase linearly with number of workers, whereas ScaleCom remains constant.

**Q6(R3) LP filter and momentum SGD; sensitivity of $\beta$ in LP filter:** [momentum SGD]: Intuitively, momentum
SGD can be viewed as a form of filtering (moving average) on current and past gradients, which smooths out noisy
gradients to update weight more accurately. Analogously, we perform filtering on the residual gradients (see eq.(5))
to improve signal integrity in local memory. Connection will be discussed in the revised version. [$\beta$ sensitivity]: We
observed that $\beta$ is robust to different networks' convergence in the range of 0.1-0.3. Thus, $\beta$ 0.1 is used in Table2.

**Q7(R4,R1) top-$k$ index commun. and sync:** (i) [commun.]: Since the index vector has the same degree of compres-
sion as the gradient vector, it occupies only 0.5% of baseline commun. time (see Figure(b) in Q5). Also, the cost
remains constant with increased workers ($\mathcal{O}(1)$ scalability) (ii) [sync]: While ScaleComp incurs an additional sync step,
it has negligible impact on performance. Similar to fully sync. SGD the slowest worker determines when the gradient
commun. can begin. Once this point is reached by all workers, additional sync for handshaking cost little extra time.

**Q8 (R4) Section 3 (theory) exposition and intuition:** We provided the following table to explain section 3's main
results and connected them to other parts of paper. For Remark 4, linear speedup refers to that when $T$ is large enough,
$1/\sqrt{nT}$ leads convergence rate. As worker number $n$ increases, required iteration $T$ linearly decreases to achieve the
same convergence[arxiv.org/abs/1705.09056]. Our theorem 1 shows this; indicates its applicability in distributed training.

| | Lemma1: contraction property | Lemma2: contraction in distributed setting | Theorem1: ScaleCom's convergence rate same as SGD ($1/\sqrt{T}$) |
|---|---|---|---|
| **Intuition** | Higher correlation between workers brings CLT-$k$ closer to true top-$k$. | Require positive correlation between workers in distr. setting | Ideally ScaleCom's noise does not impact final conv. results |
| **Connect to exp.** | Fig.2 and 3 show high correlation so our contraction is close to true top-$k$. | Fig.2 and 3 show positive correlation between workers | Table 1,2 (Fig4,5) verified ScaleCom's convergence same as baseline |

[Meta-Review · NeurIPS 2020]

This paper proposes a gradient compression technique to speed up all-reduce type gradient accumulation for optimization with large minibatches. The reviewers were positive on average (5,6,6,7) but did point out several concerns. Author response and reviewer discussion did not have a significant impact in changing the content or scores of the reviews. To me the author feedback does look to address the most serious concerns, and in addition I think the experimental validation of the proposed method looks strong. I therefore recommend accepting this paper.